# Technical traditions and individual variability in the Early Neolithic: Linear pottery culture flint knappers in the Aisne Valley (France)

**Pierre Allard**[1☯]*, **Solène Denis**[2☯]

**1** UMR 8068 Technologie et Ethnologie des Mondes Préhistoriques, CNRS-Université Paris 1- Université Paris Nanterre, Nanterre, France, **2** Department of Archaeology and Museology, Masaryk University, Brno, Czech Republic

☯ These authors contributed equally to this work.
* pierre.allard@cnrs.fr

## Abstract

For the Early Neolithic lithic industry in Western Europe (5500–4800 BCE), the study of technical behaviors, recognition of technical traditions, and even more so, idiosyncratic manifestations are not widespread. In this study, we propose an original approach to lithic industries based on the identification of "communities of practice" and individuals within housing units. The comparison of lithic series from the Meuse, Rhine and Seine basins allowed us to identify different technical traditions in the Early Neolithic. The study of three dwelling units at two sites in the Aisne Valley in France shows that it is possible to distinguish different flint blade debitages, which we interpreted as the work of different knappers. This novel study of hand-finding in the villages of the first agro-pastoralists populations proves stimulating for the renewal of perspectives on the interpretation of the organization of activities within villages.

## Introduction: From the *chaîne opératoire* to the identification of individuals

The Neolithic developed in temperate Europe with the *Rubané*, or Linear Pottery Culture (LPC). This large cultural entity diffused to the west from Transdanubia (Hungary) to the Paris Basin (France). In approximately six centuries, from 5600 to 5000 BCE, Neolithic life-ways reached most regions in northwestern Europe. In western Europe, the LPC was largely homogeneous (architecture, pottery styles, mortuary practices, etc.) except at the transition from the 6th to 5th millennium BCE when it fragmented into a mosaic of cultural entities (Fig 1). Between 4950 and 4650 BCE, much of northern France and Belgium was occupied by Neolithic villages of the Blicquy/Villeneuve-Saint-Germain (BQY/VSG) culture. In western Germany, the Hinkelstein and Grossgartach cultures, followed by the Planig-Friedberg, succeeded the LPC and are contemporary with the BQY/VSG. These various cultures are generally grouped under the term "Danubian Early Neolithic." The structure of the villages and hamlets was similar, with remarkable stability in the general architecture of the houses, which were

**Funding:** The authors received no specific funding for this work.

**Competing interests:** The authors have declared that no competing interests exist.

**Fig 1. Map of the Early Neolithic in the Rhine, Meuse and Paris basins.** The two steps of the Early Neolithic are represented by the Linear Pottery Culture (LPC or LBK) and the second stage by the Blicquy / Villeneuve-Saint-Germain culture (BVSG) in the Paris Basin and Belgium and in western Germany, the Hinkelstein and Grossgartach cultures, followed by the Planig-Friedberg. Bacground map: Digital elevation model BD ALTI® from the IGN, 16 of february 2022.

bordered by lateral pits containing domestic activity waste. Therefore, none of the archaeological assemblages originate from a level or remnant of one that can be considered as an occupation layer of this period.

In this article, we propose a new approach to Danubian lithic industries based on identifying "communities of practice" and individuals within the LPC households.

Many studies of the lithic industries of the first agro-pastoral communities in temperate Europe rely on analyses of raw materials and their circulation networks to understand the economic organization, contacts, and exchanges between communities (e.g., [1]). Studies that report technical or stylistic variability in flint blade production are less frequent [2–6]. At the scale of our study region (the Rhine/Meuse/Seine basins), it appears that a chronological evolution of blade manufacturing methods is thus perceptible. In the earliest LPC (*älteste Bandkeramik*) lithic industry, D. Gronenborn reports the coexistence of what he calls two technical "traditions" for blade extraction [4]. The first produced narrow regular blades with a large facetted platform and seems to correspond to the debitage in the Early LPC zone. The second concerns products with a small plain platform, tending toward punctiform, which seems to be associated with the local Mesolithic. In the same zone, from the next stage (LPC II) to the final one, we see longer and less regular products with wide, slightly prepared platforms. These pieces, identified in northern Rhineland and the Netherlands [1,7], correspond to the modalities described for the Hesbaye. The two "traditions" mentioned in the early stage thus appear to disappear rapidly in the zone concerned. In all studies, the qualifications of these debitage types have been vague, ranging from a true "tradition" for D. Gronenborn, to the notion of

styles or variants. In contrast to pottery studies (e.g., [8]), the interpretative framework from which to investigate the informative potential of these observations remains to be established.

In studies of prehistoric artifact manufacturing, the technological and *chaîne opératoire* (CO) approaches can be used together to analyze the technological procedures and knowledge obtained via cultural transmission, thus reflecting the society in which they existed [9–14]. Most researchers define the CO as a series of operations that transform a raw material into a finished product [15]. The constraints associated with the raw materials and cultural factors are responsible for the variability observed in its implementation, thus resulting in different "ways of doing" and the ensuing "traditions" of distinct social groups, whatever their nature [16]. Because COs are a product of the skills acquired by individuals via apprenticeship within social groups governed by cultural rules (e.g., [17]), they rapidly appear as expressions of cultural traditions (e.g., [10]). However, apprenticeship is structured by cultural rules in the sense that the individuals belong to social groups. Due to the constraints that influence the learning processes, varying technical practices are associated with distinct "communities of practice," a concept defined by Lave and Wenger [18] to describe social groups that share the "same way of doing things" [19]. In this way, technical traditions represent social groups more than morphological or stylistic features (e.g., [19]). In ceramic studies, actualistic studies have revealed regularities that associate the variability of COs with social entities, whose meaning is understood via several theoretical frameworks [8,16,20]. The technological approach to lithic artifacts, in France, for example, where a distinction is made between technique and method [21,22], reveals the importance attributed to the project and the abilities of the actor. The intention is perceived as highly constrained by the technical traditions of the group, which are transmitted via apprenticeship. Therefore, the CO concept is explicitly enriched by the concept of "technological practice," defined as the COs and competencies implicated in manufacturing objects that fulfill socio-economic needs [19].

In summary, to understand the variability of material culture, it is necessary to comprehend the factors that influence technological practices. The nature of a prehistoric artifact is determined by the technical, economic, and social choices dictated by cultural traditions [23]. At the individual level, technological analysis, faced with behavioral variability, enables one to identify recurring responses to certain situations at the individual level. For example, regular knapping practice enables an artisan to overcome circumstantial events, which then no longer hinder the completion of the project. When confronted with similar situations, the individual tends to repeat the operational and/or conceptual responses they know. The conjugation of knowledge, motor skills, and psychological mastery can thus indicate individual variability within a general scheme shared by the community [24].

Therefore, the question of transmission, and thus of apprenticeship and skill levels, is essential to identifying idiosyncrasies. Many theoretical and conceptional studies have contributed to this question [cf. history in 25]. In lithic analyses, these studies often focus on the Paleolithic, including a few major studies of blade manufacturing, such as the analyses of the Magdalenian at Etiolles and Pincevent [24,26] and, less frequently, on unique bladelet productions [27].

The contribution of flintknapping experiments is well established. Modern flint knappers contribute to the understanding of different aspects such as the stages of the *chaîne opératoire*, the knowledge involved, the knapping tools, but also the skill or the recognition of the levels of know-how [28]. Through our understanding of the different choices and actions on the material, both conscious and unconscious, it becomes possible to also understand the cognitive processes of the prehistoric knapper [14]. Experimental studies thus serve as a basis for demonstrating the relevance of tracking individual signatures. Different methods and criteria have been used in this demonstration, depending on the productions studied. Three important elements can be underlined in this effort to distinguish individuals: the highlighting of

different levels of know-how; the identification of different registers of knowledge or the presence of particular processes. The identification of different levels of know-how within prehistoric productions is a subject that has been extensively dealt with in experiments, allowing the establishment of criteria used in the study of the archaeological material. These are technical criteria linked to the motor skills of the knapper and his ability to adapt to the volume and the quality of the raw material exploited [27,29,30]. For bifacial productions where the search for symmetry is crucial, statistical analysis can be used to distinguish individuals, based on the metric criteria of the objects. Experiments on Pueblos points or Danish Bronze Age conducted by John Whittaker [31] and Joel Gunn [32] are two examples of flintknapping experiments where the aim was to identify different knappers on the basis of the standardisation of production, essentially linked to the level of skill of the knappers. Thanks to the detailed analysis of the knapping gestures, it was sometimes possible to demonstrate that certain individuals, for the same production, call upon different registers of knowledge. For example, for the Middle Stone Age, the detailed analysis of the methods of detachment of the preparatory removals of a Levallois debitage sequence (succession and position of the removals), carried out by three modern knappers, made it possible to distinguish them [33]. The experiments carried out by Jacques Pelegrin to study productivity in the contexts of axe production workshops are distinguished from the archaeological material by a clearly more assertive care in the shaping, which the author was unable to dispense with in order to stick to the archaeological productions [34]. Finally, particularly for the production of blades, a number of experiments have attempted to undertake productivity studies in the context of production workshops or extraction sites [35,36] and to establish criteria for distinguishing percussion techniques [37,38]. It is on this occasion that some studies have highlighted the existence of processes that could be interpreted as individual signatures rather than the percussion technique. The study by Driscoll and Garcia emphasizes, in this perspective, that lipping and platform preparation and crushing are more a reflection of idiosyncratic manifestations. Finally, a detailed analysis of the tools and gestures of percussion on a microscopic scale underlines the as yet little exploited potential of use-wear analyses in this perspective of distinguishing individuals [39]. Thus, the theoretical framework and its application to archaeological collections have demonstrated their potential to track individual signatures. But analyses of technical behaviors, the identification of technical traditions, and, even more so, idiosyncratic manifestations are very rare in studies of Early Neolithic LPC lithic industries [40]. Before continuing, we should point out that identifications of variability in lithic tool manufacturing COs most often involve blade productions because they require a long and intensive apprenticeship [41,42], which is associated with visible features that reflect technical regularities.

This article begins with a thorough methodological review of the range of technological data available for the blade productions of the first agro-pastoral populations in the Seine/Meuse basins, the region of the terminal expansion of the Early Neolithic in temperate Europe. We aim to demonstrate the relevance of a technological analysis method that contributes to interpreting the variability of lithic artifacts both in terms of communities of practice and individuals.

## Materials and methods

### Materials

The flint artifacts involved in this study originate from pits bordering the lateral walls of habitation units in a generally detrital context. These lithic objects were thus mixed and found in association (or not) with other artifact categories (ceramic, fauna, etc.) in a secondary deposit [43,44]. In contrast to central Europe (e.g., [45]), the artifacts recovered in lateral pits in the

Seine/Meuse sector reflect the activities of the houses next to them, and the pits seem to have been filled rapidly, within three to five years [43].

Our analysis is structured according to three complementary data categories:

i.  first, the general features of the lithic industries in our study zone are drawn from the doctoral research of the authors:

  - for the LPC, 15 sites, approximately 55000 pieces [2].

  - for the BQY/VSG, 11 sites, approximately 45500 pieces [46].

ii.  second, the criteria used to identify communities of practice are defined based on the analysis of 1941 pieces from the LPC and BQY/VSG sites of Vaux-et-Borset (Hesbaye, Belgium [47]);

iii.  finally, our analysis of the idiosyncratic data is part of our continuing research as part of the ANR Homes program (dir. C. Hamon 2020–2024), whose preliminary results we present here. The collections studied in this program originate from LPC sites in the Aisne Valley and are all dated to the end of the LPC sequence [48]. The research project, which was initiated in the Aisne River valley almost 40 years ago, is based on systematic monitoring of gravel extraction along an 80 km stretch of the middle Aisne valley, combined with rescue excavations [48]. Thus, 15 LPC sites were excavated, with over 90 house plans and almost 85 graves, including the major site of Cuiry-làs-Chaudardes, which has the longest duration of occupation and the largest number of domestic units recorded in the Aisne valley [48]. The LPC sequence in the Aisne valley corresponds to the final LPC of Central Europe, with C14 dates for the sites falling between 5100 and 4900 BCE. A major advantage of this region is the relatively clear layout of settlements, with very few overlapping house units, due to the short duration of the local LPC sequence. Therefore, the artefacts found in the lateral pits on settlement sites can easily be attributed to a given house unit. Despite the quantity of finds (17 000 flints for the 15 sites), cores are rare and there are only rare informative refitting. Blade debitage seems homogeneous in the series studied. The aim is to obtain products with parallel, rectilinear edges. The estimated dimension range is 8–10 cm in length, 1,4–2 cm in width and 2–6 mm in thickness. Toolkits are stereotyped since the series from all the sites are almost entirely made up of scrapers, splintered pieces, retouched blades and flakes, arrowheads, sickle blades and burins. It is the study of this region that has made it possible to specify the blade debitage of the Seine Basin LPC, which is referred to as Alpha in this article. While there is no raw material economy stricto sensu (a material used exclusively for one type of production), there are clear preferences for high-quality regional flints (especially, Senonian flint) for blade production and for local materials for expedient productions [2,49]. At this stage of our research, we believe that flint blades were made in all domestic units. Our technological analysis shows that flint knapping waste products are present in all of the houses, including those with few remains.

We more specifically selected two LPC sites in the Aisne Valley: Bucy-le-Long "La Héronnière" (BLH) and Cuiry-lès-Chaudardes "Les Fontinettes" (CCF) excavated by the UMR 8215 Trajectoires laboratory [48]. We chose these sites because their well-preserved houses had a sufficient number of blades for our type of analysis, enabling us to record the features of more than 20 proximal blade pieces. The houses selected for this study are among the richest and best dated in the region and have been described elsewhere [2]. House 380 contains 1458 pieces including 90 blade tools and 3 cores, house 570 has 190 pieces including 48 blades tools and no core, house 120 had 434 pieces including 48 blade tools and no core. The composition

**Table 1. Number of blades with butt selected.**

| Houses | BLH 120 | | CCF 380 | | CCF 570 | | total |
|---|---|---|---|---|---|---|---|
| *pits* | *122* | *124* | *378* | *382* | *556* | *598* | |
| nb pieces | 7 | 16 | 33 | 14 | 26 | 4 | 100 |
| total nb | 23 | | 47 | | 30 | | |

of the toolkit is strictly the same. The tools of the knappers, hammerstones or punches, are totally absent from the sites.

The LPC sites in the Aisne Valley are usually very poor, with an average of only 150 lithic artifacts per building. At Bucy-le-Long "La Héronnière" (BLH 120), only one house yielded a sufficient number of blades for our analysis. At Cuiry-lès-Chaudardes, we selected, for a first test, two buildings (CCF 380 and CCF 570) among those that yielded the highest number of lithic artifacts.

The chronological seriation of the dwelling units in the Aisne valley is determined by a stylistic analysis of the decorated ceramics. Several recent studies have made it possible to specify the seriation of the region and to integrate it into the general chronology of the Western European LPC [50–52]. In the Aisne valley, statistical analyses of the types of decorations allow to distinguish four stages. House 120 of Bucy-le-Long "La Héronnière" is attributed to the first one [51,53]. Houses 380 and 570 from Cuiry-lès-Chaudardes are both dated to the second stage of the Aisne seriation [52].

These three houses (two at CCF and one at BLH) yielded at least 100 blades and blade tools. After sorting the proximal ends of these blades and eliminating those with damaged or retouched platforms, and those with no platform, we were able to study 100 pieces, or 23 to 47 pieces per dwelling unit (Table 1).

This latter dataset thus enabled us to make very detailed technological observations aimed at detecting distinctive features. The two other datasets provided a frame of reference from which to develop our interpretations.

## Method

The absence or scarcity of refitting and the secondary detrital nature of the sedimentary contexts required us to work with a simple method based on descriptions of *schémas diacritiques* (diacritic diagrams) of the lithic artifacts. A diacritic diagram is a schematic representation of an object that enables one, with a small number of graphic means, to show the order and arrangement of the last operations involved in its manufacturing [22]. Using this method, it is possible to apprehend fragments of a lithic manufacturing sequence and reconstruct their order in the manufacturing process, enabling an identification of the flaking method used [54]. The constraints of this method are still significant, however, because, as J. Pelegrin explains: "Mental refitting gives an averaged view of reality that lacks specific cases or, via summation, amalgamates specific modalities" [54]. The broader trends thus blur our identification and understanding of the individual modalities chosen by individual knappers.

A knapping technique is identified based on macromorphological criteria, the stigmata revealed through experimentation by modern knappers and, by analogy, the stigmata studied and identified in prehistoric lithic assemblages [12,55–60]. Concerning indirect percussion, we can refer to the extensive research of J. Pelegrin on both the history of research and the diagnostic criteria identified for this knapping technique [41,57,59,61–63]. Based on the knowledge already obtained by the experimenters of this percussion technique, we know that "punch" flaking is associated with cores having a plain striking platform and a "comfortable"

platform angle of 80 to 90˚. The platform edge asperities can be partially abraded or not at all because the punch tool ensures the precision of the impact zone. The punch technique enables the production of a high number of regular blades. The pronounced concavity of the blade butt, its external overhanging edge, or the presence of a half-circle behind the platform (sometimes called a "half-moon") are precious indicators enabling the identification of this percussion technique (Fig 2). During the Danubian period, the punch appears to have been the only knapping tool used for blade debitage, except at the very end of the sequence when it seems that the hard stone hammer technique was sometimes used, but we will not address this latter technique here. The foundation of our analysis of our first dataset is the reconstruction of debitage methods and techniques.

In the analysis of our other two datasets, we focus on the finished and semi-finished products, meaning the blades. The criteria we consider are the dimensions and characteristics of the laminar products and their impact zone:

- blade dimensions and their position in the *chaîne opératoire*;

- blade section and blade detachment order;

- butt types reflecting the techniques used to prepare and maintain them;

- striking platform dimensions;

- the degree of preparation of the platform edge and qualitative description of their preparation techniques, and;

- evaluation of the platform angle.

These technical criteria enable us to identify the different actions and tools used, and by extension, to identify communities of practice. But based on these criteria, how can we distinguish individuals? Even if this field is still exploratory for this period, the structuration of sites into distinct habitation units offers a relevant analysis framework, especially since LPC productions are considered to be domestic [2,64].

## Results

### Method and technique: Identifying debitage styles

There are large quantities of small, straight blades with parallel edges in the LPC, made with the indirect percussion technique. The knappers prepared the cores in two stages separated by a change in their percussion tool. In the first stage, they used direct hard stone percussion to rough out the core and prepare the general orientation of the volume to be knapped. In the second stage, they used indirect punch percussion to complete the core preparation by creating longitudinal crests. In the Hesbaye LPC productions (Belgium), the last step in preparing a core was to open the striking platform, and this was done after the crests were created [2,65]. In other regions, however, there is little evidence of striking platform creation. Everywhere, the creation of a single crest is the simplest and most frequent debitage version. In this case, the flakes detached to prepare the core sides and back remained cortical, but multiple crest preparations have been recorded in most regions (Hesbaye, Hainaut, Paris Basin). Except in the core preparations with several crests, the core sides are only slightly prepared and gradually modified during the flaking process by various detachments starting from the platform (laminar flakes, bladelets, and cortical blades). Axial laminar detachments are used to maintain the flaking surface. The flaking is generally unidirectional. Bidirectional products exist in several sites, but diacritical diagrams indicate that alternative debitage did not exist [2]. Bidirectional debitage corresponds to successive unidirectional detachments or an operation to maintain

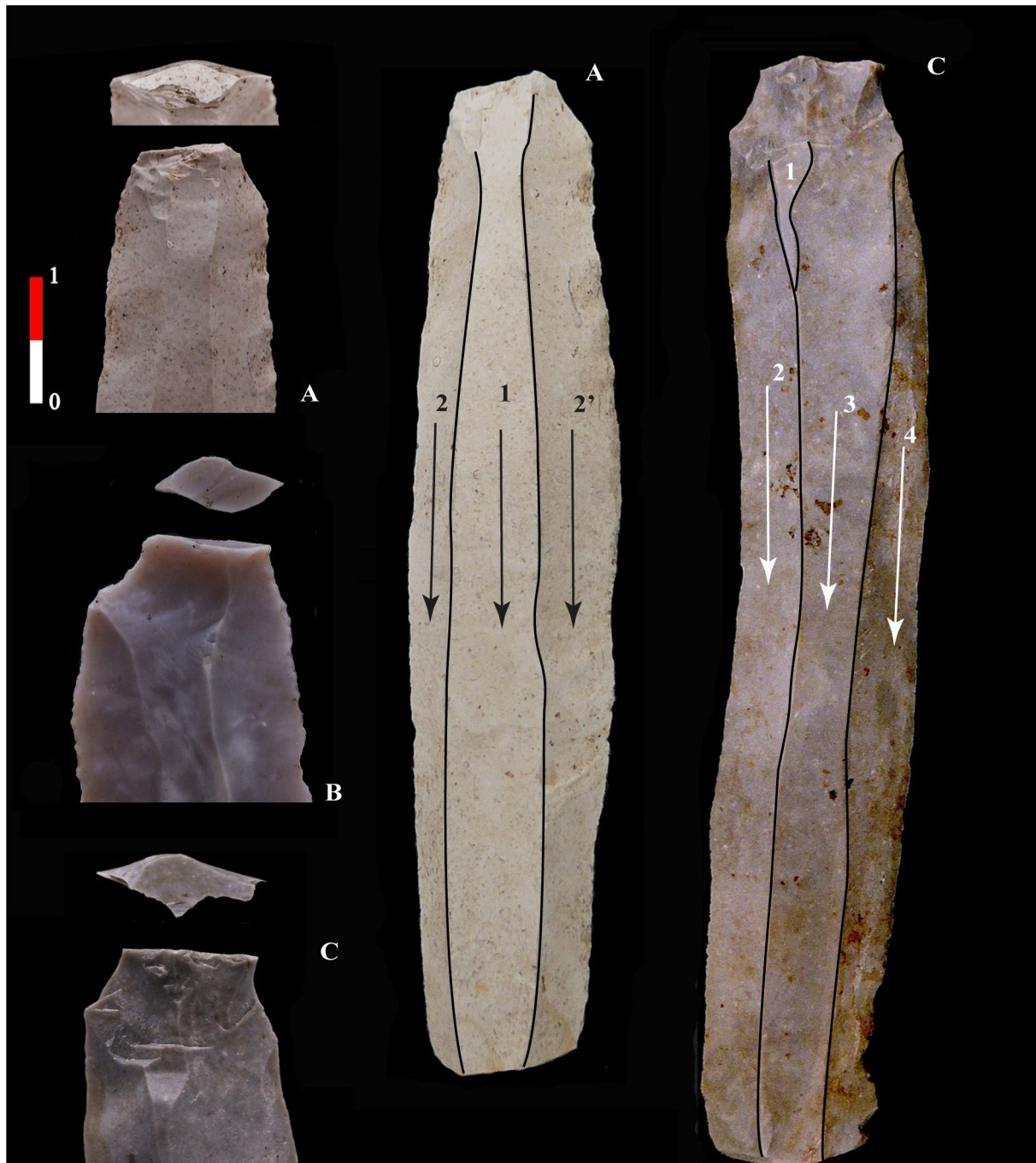

**Fig 2. Example of diagnostic blades for indirect percussion.** Three examples of blades characteristic of punch technique: A, a regular blade from Bucy-le-Long "la Fosse Tounise" (with a partially prepared overhang and a 90˚ platform angle ((BLF, 44–1280, Soissons, France); B, a blade from Bucy-le-Long "la Héronnière" with a flat butt, a half-circle behind the platform and a rough overhang (BLH, 124–2727, Soissons, France); C, typical Early Neolithic blade from Hesbaye. The butts of the blade is plain and flat, with residues of the overhang (ACM, 150–8497, Aubechies, Belgium).

the flaking surface by opening a platform on the opposite end. The base of the core was regularly maintained by distal neo-crests or partial re-preparations with the aim of recreating a suitable longitudinal convexity. The cores thus have a "pyramidal" shape with a cordiform and

curved laminar surface, which is never quadrangular. The blades have small homogeneous dimensions, from 8 to 12 cm long.

Within this standard procedure, a clear distinction exists between the Hesbaye and Seine Basin sites (mainly recorded in the Aisne Valley, in the Yonne, and at the site of Bréviandes on the Troyes plain).

The usual striking platforms of blade cores in Hesbaye are plain and flat, created by removing the top of a nodule and maintained by detaching thick tablets, characteristic of "Omalian" assemblages [66,67]. The butts of the laminar products are plain and flat with no specific modifications, and with no or little removal of the protruding dorsal edge (overhang). The punch is placed behind the edge of the striking platform, which produces blades with typical wide platforms with overhangs or residues of them (Fig 2), leading D. Cahen to propose the use of indirect percussion [65].

In the Aisne Valley LPC assemblages, the installation and opening of the striking platforms are poorly known, but the core rejuvenation method has been thoroughly studied [2]. The striking platform is regularly maintained by removing centripetal flakes with centimetric dimensions, modifying the edge of the platform and creating negative bulb scars on the upper part of the flaking surface [68]. This systematic procedure is associated with partial repairs of the platform and rare complete tablets when the multitude of flake detachments created a central dome-shaped convexity. The blade detachment is carefully prepared by partial or total abrasion of the platform edge. The butts are thinner than those of the blades from Hesbaye sites, sometimes even linear. The detachment of small flakes creates closely-spaced concavities that result in frequent "morphologically dihedral" (false-dihedral) or plain concave butts. These butts result from repairs to the striking platform rather than from an intentional preparation of the impact zone (Fig 3).

The second distinction between these two debitage types concerns flakes detached from the axial and lateral sides of the core. The Omalian debitage products, for example, include large flakes associated with axial and lateral repairs, detached by direct stone hammer percussion, attesting to the total or partial core repairs typical of this type of debitage.

In the Hainaut region, these two debitage types coexist between the center of the Paris Basin and the Hesbaye and have been named "styles" or "variants." They are interpreted as a response to the differential access to raw material sources because the material used for these variants were not strictly comparable [2].

## Chronological depth to understand the transmission mechanisms: Toward a social interpretation of technical diversity

The study of Blicquy Group sites in Belgium contributed significant results concerning the distinctions of different debitage types. For instance, the synthesis of the Blicquy Group lithic industry demonstrated the long duration of the two variants observed during the LPC, which were thus interpreted as technical traditions [46]. The site of Darion in Hesbaye yielded a few blades with specific preparation methods that were not identified for the two variants previously described. But the small size of this assemblage prevented us from making any further interpretations. However, the study of the sub-contemporary sites of Grossgartach and Planig-Friedberg in the North Rhine area confirmed the interest in searching for discrete technical features to define variants in Danubian blade debitage, and a new "way of doing" was indeed revealed [69,70]. In this case, the blade butts are dihedral and reflect a striking platform preparation procedure involving the detachment of small flakes. Above the future detached blade, a dihedral ridge corresponding to the convexity at the junction of these two flakes was formed. The punch was then placed on this dihedral ridge and may have facilitated the initiation detachment of these small, 6–8 cm long.

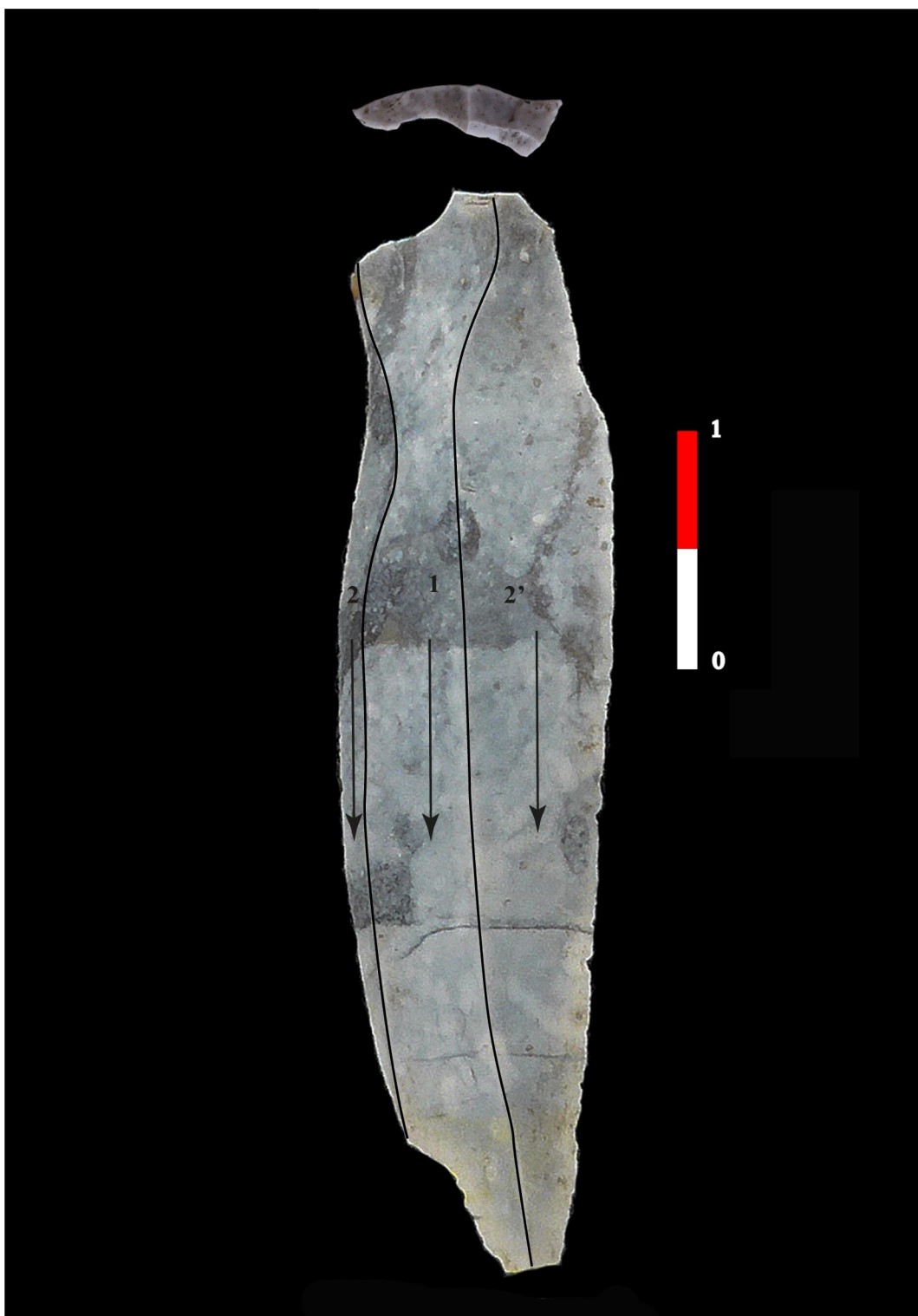

**Fig 3. Blade with a morphologically dihedral butt.** A blade from Berry-au-Bac "le Vieux Tordoir" with a "morphologically dihedral" butt, due to the detachment of small flakes for the rejuvenation of the striking platform (BVT, 635–8524, Soissons, France).

Furthermore, the debitage is frequently guided by a succession of opposed striking platforms, the core being turned over several times as the flaking progresses. In their final state, the cores are rather flat, with a preferential flaking surface on one of the wide faces. A recent analysis of eastern Belgium corpus identifies different ways of doing in the Danubian laminar debitage. Analysis of the striking platform maintenance and blade detachment procedures at the site of Vaux-et-Borset in Hesbaye indeed contributed to the distinction of five ways of doing [47,71]. This site was successively occupied during the two stages of the Early Neolithic —the LPC and the BQY/VSG—and is thus particularly relevant to the study of technical traditions. In the LPC occupation, one way of doing clearly dominates the assemblage and is identical to the way of doing identified elsewhere in the region and described before (Omalian debitage).

Conversely, during the Blicquian, this same tradition continues but coexists with three other ways of doing (Fig 4), one of which corresponds to a debitage type identified in the Paris Basin and probably exclusively represented in Hainaut during the BQY/VSG [46]. The two other ways of doing remain to be clarified.

One seems to be a hybrid of the two preceding ones, showing intensive interactions between knappers [47]. The other one shares some aspects with the tradition reported in Germany, even if some criteria have not been identified [71].

One of the laminar productions' main technical variabilities thus lies in the treatment of the striking platform and the blade detachment procedure.

In the Blicquy/Villeneuve-Saint-Germain context, identifying the persistence of LPC debitage "styles" sheds new light on the subject [46]. This transmission between generations is a key argument for their interpretation as a reflection of technical traditions or communities of practice. Moreover, these styles can no longer be correlated with the availability of siliceous resources. The new discovery of the LPC site of Bréviandes on the Troyes plain and the synthesis of the VSG sites in the Marne Valley, two regions with abundant raw materials, show that the laminar production procedure corresponds to the features identified for the Paris Basin "style" [40,72], initially considered as less expensive in terms of raw materials.

The technological analysis of Danubian industries thus enabled the identification of debitage types that reflect different ways of doing associated with a given social group. This interpretation in no way negates the idea that the observed technical variability could have emerged as an adaptation to the available resources. These ways of doing are transmitted by apprenticeship and thus reflect "apprenticeship lineages." The use of the term "technical traditions" is acceptable when the vertical transmission of a way of doing can be observed, as is the case for the Paris Basin traditions (the "Alpha" tradition after [47]) and the Hesbaye ("Beta" tradition). Lacking systematic conjoins and due to the often-incomplete representation of the stages of the debitage *chaîne opératoire* at the sites, the blades with a preserved butt provide key information for the distinction of these ways of doing [13]. In effect, we can observe both the treatment of the striking platform and the manner of preparing the external platform edge before the blades are detached. These two parameters imply actions and tools whose combination enable the identification of variants in the laminar debitage *chaîne opératoires*, interpreted as a reflection of these communities of practice. Alpha tradition will be abundantly mentioned hereafter. To sum up its technical description, we can underline that the straight blades with parallel edges from the Alpha tradition are obtained by indirect percussion technique. The knappers prepared the cores in two stages separated by a change in their percussion tool. In the first stage, they used direct hard stone percussion to rough out the core and prepare the general orientation of the volume to be knapped. In the second stage, they used indirect punch percussion to complete the core preparation by creating longitudinal crests. The flaking is unidirectional. The cores thus have a "pyramidal" shape with a cordiform and curved laminar

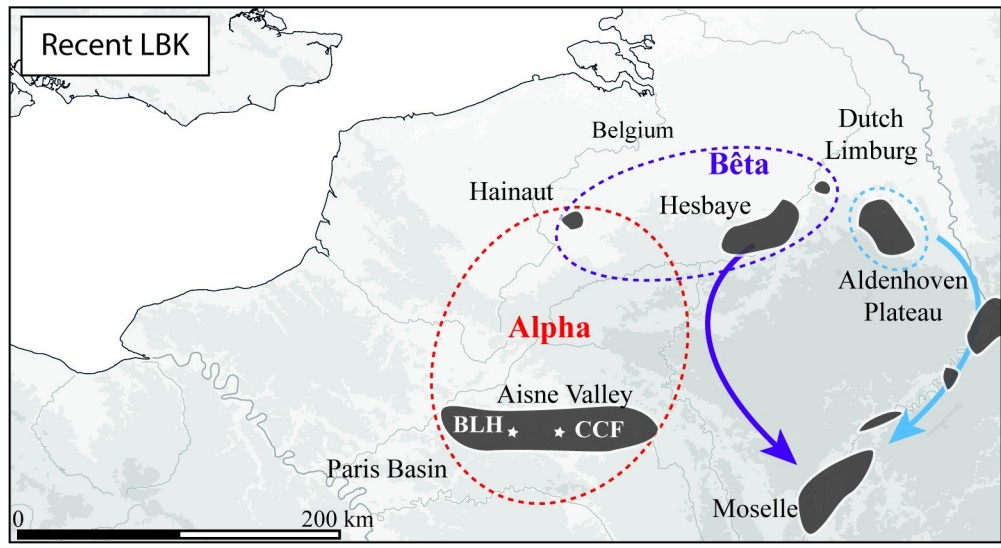

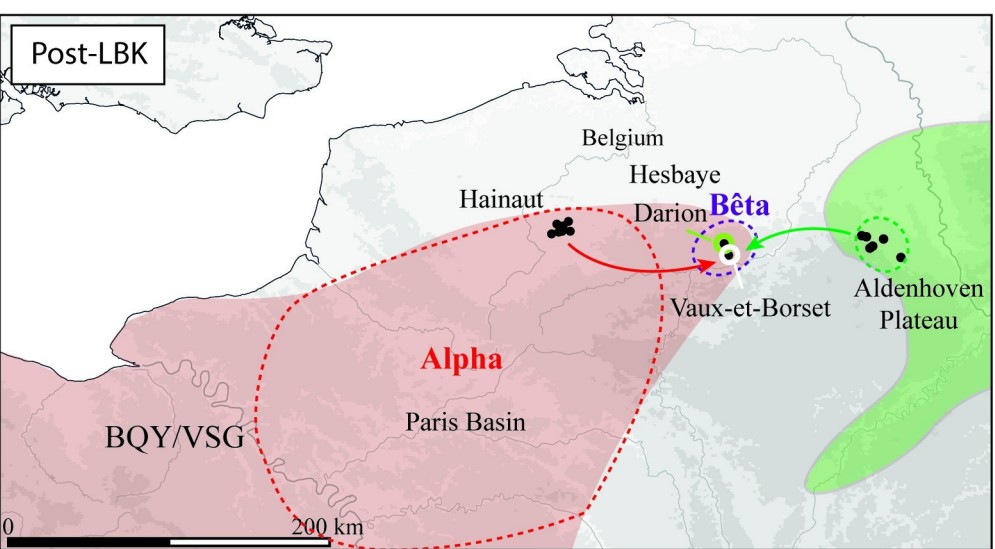

**Fig 4. Map of the different traditions of blade debitage.** The map shows the distribution of the main flint knapping traditions between the late LPC and post-LPC. During the first stage, strong links exist through the flint blade circulation networks Map base: Digital elevation model BD ALTI® from the IGN, 16 of february 2022.

surface. The *debitage* is almost rotating. The striking platform is regularly maintained by removing centripetal flakes with centimetric dimensions, modifying the edge of the platform and creating negative bulb scars on the upper part of the flaking surface. This systematic procedure is associated with partial repairs of the platform and rare complete tablets when the multitude of flake detachments created a central dome-shaped convexity. The blade detachment is carefully prepared by abrasion of the overhang made with a stone tool. The detachment of small flakes creates closely-spaced concavities that result in frequent "morphologically dihedral" (false-dihedral) or plain concave butts. These butts result from repairs to the striking platform rather than from an intentional preparation of the impact zone. Knappers from A tradition seem to know the specific arrangements allowing the recurring production of blades with a regular trapezoidal cross-section.

## Domestic units in the Aisne Valley: Revealing individual variability

Based on our third dataset, we aimed to reveal the variability present at the domestic scale. The lithic industry of LPC sites in the Aisne Valley was predominantly made on four flint varieties. The Aisne valley is set in a geological landscape that offers many easily accessible silicites, but at varying distances from the immediate surroundings of the sites to one- or two-days walking. On a wider scale (20–50 km), the flint resources are derived from primary formations of Tertiary (Bartonian and Ludian) and Senonian (Campanian) levels toward the south, Turonian flint in north-eastern direction and Campanian levels of the Oise in northwestern direction. All these deposits are on average distant from 30 to 50 km away from the villages. Locally, the alluvium includes small blocks Turonian flints from the Ardennes, greenish cortical blocks from the Thanetian, undifferentiated Tertiary silicites, and quartzite sandstone. All the LBK sites in the region systematically present these different materials [73] (Table 2). Our previous work shows that they were exploited for the production of blades and that there is no economy of raw materials in the strict sense [74]. It means that despite the material diversity, knapping objectives convey toward the same blade production to obtain the same toolkit. Despite the preferential choice oriented towards good quality regional flints, the production of blades is similar from one material to the another [2]. In fact, there are two technical arguments that can explain the exploitation of different sources. On the one hand the quality of these different flints is comparable. The homogeneity of the matrices is comparable. The only difference is lying in the morphologies of the blocks (tabular for the tertiary flint versus nodules). On the other hand, the production aims at small size blades (8 to 12 cm), which allows to exploit small volumes, easy to find in these different deposits of good quality for knapping.

The first criteria considered to study the variability at the domestic scale is the morphometry of the blade butts. We identified a majority of plain butts (Table 3), while concave plain butts and residual dihedral butts (false-dihedrals) are also frequent. The three other types (technical dihedral, linear, punctiform) are much less common. This broadly represents the debitage method identified in this region and is associated with the Alpha community of practice. However, the distribution of types per house shows a clear difference between the two Cuiry-lès-Chaudardes units and clearly distinguishes the Bucy-le-Long house (Fig 5). No other concave butt was observed there and false-dihedrals are rare, while linear and punctiform butts are numerous here but nearly absent at Cuiry-lès-Chaudardes. These elements constitute a significant point of divergence from the technical characteristic of the Alpha tradition, suggesting the existence of knappers using another way of doing at this BLH habitation.

This histogram shows an important difference in the frequency of butt types between the house in Bucy-le-Long "La Heronnière" and the two in Cuiry-lès-Chaudardes.

In the village of Cuiry-lès-Chaudardes, House 570 is somewhat distinct in its low proportion of plain butts and higher number of concave and false-dihedral ones.

The butt dimensions seem to be more relevant in distinguishing between the two habitation units at Cuiry (Table 4). The average length and thickness of the different butt types show perceptible variation, especially in House 380 at CCF, which displays wider and thicker butts.

**Table 2. The raw materials of the selected blades.**

| Raw material | BLH 120 | CCF 380 | CCF 570 | total |
|:---:|:---:|:---:|:---:|:---:|
| Senonian | 12 | 11 | 23 | 46 |
| Bartonian | 6 | 32 | 7 | 45 |
| Turonian | 2 | 2 | - | 4 |
| Quartzite | 2 | 1 | - | 3 |
| Other | 1 | 1 | - | 2 |
| total | 23 | 47 | 30 | 100 |

**Table 3. Distribution of types of butt per house.**

| butts | BLH 120 | CCF 380 | CCF 570 | Total |
|---|---|---|---|---|
| flat | 11 | 33 | 16 | 60 |
| concave | - | 6 | 8 | 14 |
| false-dihedral | 1 | 5 | 5 | 11 |
| dihedral | 2 | - | - | 2 |
| linear | 6 | 1 | - | 7 |
| punctiform | 2 | 2 | 1 | 5 |
| indert. | 1 | - | - | 1 |
| Total | 23 | 47 | 30 | 100 |

This observation can be seen in the spread of the widths and thicknesses of these butts. House 570 is distinguished by a slight difference in these measures: nearly 80% of the butts are 4 to 8 mm wide and 69% between 2 and 4 mm thick. Concretely, the blade butts of House 570 are much smaller than those of House 380.

Our analysis of the butt characteristics and dimensions shows two forms of variability:

- between the two villages studied, comprised of the morphology of butts atypical for the BLH, and;

- between the habitation units of the village of Cuiry-lès-Chaudardes, essentially comprised of variability in the butt dimensions.

But it is possible to further this analysis of technical variability via an analysis of the preparations of the edge of the striking platform.

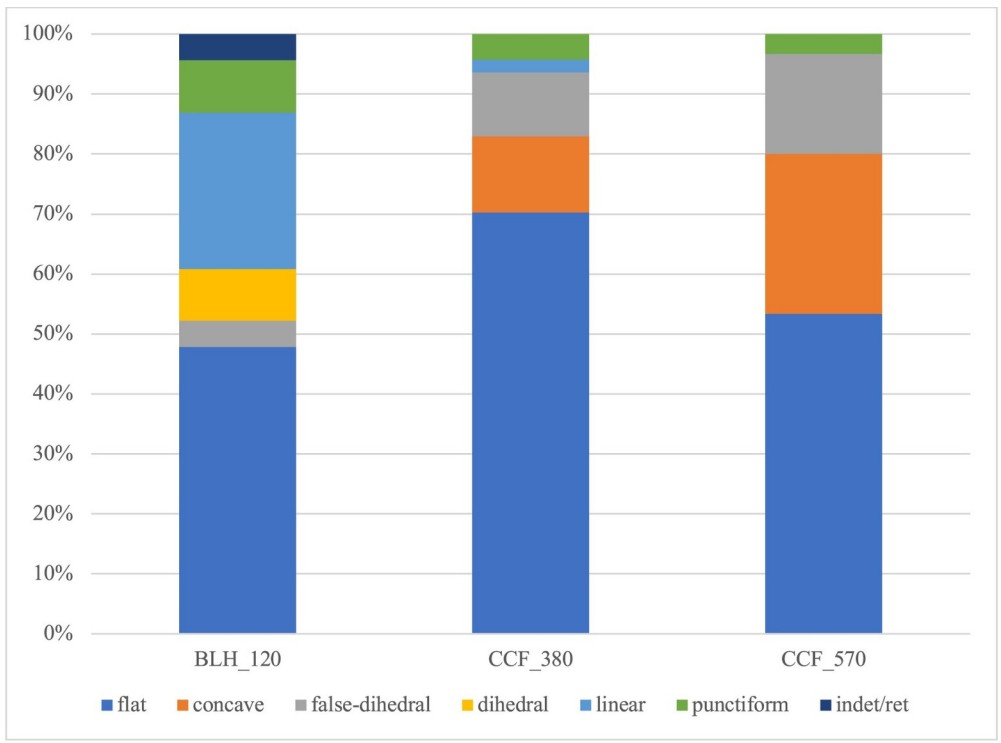

**Fig 5. Frequency of butt types per house.**

**Table 4. Average width and thickness dimensions by type of butt per house.**

|  | BLH 120 | | CCF 380 | | CCF 570 | |
|---|---|---|---|---|---|---|
|  | av.width | av. thick. | av.width | av. thick. | av.width | av. thick. |
| plain | 8.2 | 2.9 | 9.5 | 4.0 | 6.1 | 2.6 |
| concave | - | - | 8.2 | 3.6 | 7.1 | 3.4 |
| false-dihedral | 11.7 | 2.6 | 14.4 | 6.8 | 8.4 | 3.3 |
| dihedral | 6.3 | 1.9 | - | - | - | - |
| linear | 5.6 | 1.2 | 6.1 | 2.1 | - | - |
| punctiform | 2 | 1.1 | 3.7 | 1.9 | 3.5 | 1.8 |

A second essential criterion in distinguishing components within the habitation units focuses on the actions linked to the butt overhang preparation. Firstly, this consists of a simple description that enabled us to distinguish three degrees of butt overhang preparation: no preparation; partial preparation, and total preparation. This preparation is systematically carried out with a stone tool characteristic of the Alpha tradition.

Firstly, a basic comparison of the butt overhang preparations superimposes the preceding observations: House 120 at BLH is clearly distinct from CCF by its blades with systematically and often intensively prepared overhangs. The degree of overhang preparation is similar between the two Cuiry-lès-Chaudardes houses, and the blades of House 380 are more often prepared than those of House 570. This is relevant because it attests to the independence of the criteria of butt dimensions and the degree of overhang preparation at this stage of research. Even if they are more frequently prepared in House 380, the butt dimensions remain greater than those of House 570, adding to the criteria of distinction between these two entities.

Secondly, this work already shows the existence of several components within the habitation units. On this issue, Table 5 is very demonstrative. Two preparation modes are distinguished in House 120 at BLH via the coexistence of blades with partially prepared plain butts and blades with totally prepared linear butts. This double component is clearly visible on the diagram (Fig 6). In this latter group, the totally prepared butts are very narrow and from 1 to 2 mm thick, in contrast to the other set in which all the butts are more than 2.5 m thick. There is a clear separation between the group of totally prepared butts and the group comprised of untreated or partially prepared butts (Table 5). A difference of a nearly two times the width separates the two groups in House 120 of BLH (5.64 to 10.38 mm). This difference is even more significant for the thicknesses (1.55 to 5.06 mm).

For House 380 at CCF, the butt dimensions have much larger width and thickness ranges than the other houses, and we can distinguish two groups with a limit of 8 mm wide and 4 mm thick (Fig 6). The first group includes almost all the butts with a totally abraded overhang and a few partially prepared butts, while the second group comprises untreated and partially prepared butts. As for BLH, a width of nearly 3 mm distinguishes this latter group form the other one, and an average of 1 mm in thickness (Table 5). Therefore, by coupling the butt dimensions and degree of overhang preparations, it is, like for BLH, possible to distinguish a double

**Table 5. Average butt width and thickness by type of overhang preparation.**

| Sites | untreated overhang | | | partially prepared overhang | | | totally prepared overhang | | |
|---|---|---|---|---|---|---|---|---|---|
| butt size | average width | average thickness | nb of pieces | average width | average thickness | nb of pieces | average width | average thickness | nb of pieces |
| CCF 380 | 10.89 | 5.39 | 11 | 10.26 | 3.98 | 13 | 7.76 | 2.82 | 19 |
| CCF 570 | 6.42 | 2.56 | 4 | 7.23 | 3 | 5 | 6.69 | 3.14 | 15 |
| BLH 120 | 10.38 | 5.06 | 1 | 8.99 | 3.22 | 7 | 5.64 | 1.55 | 15 |

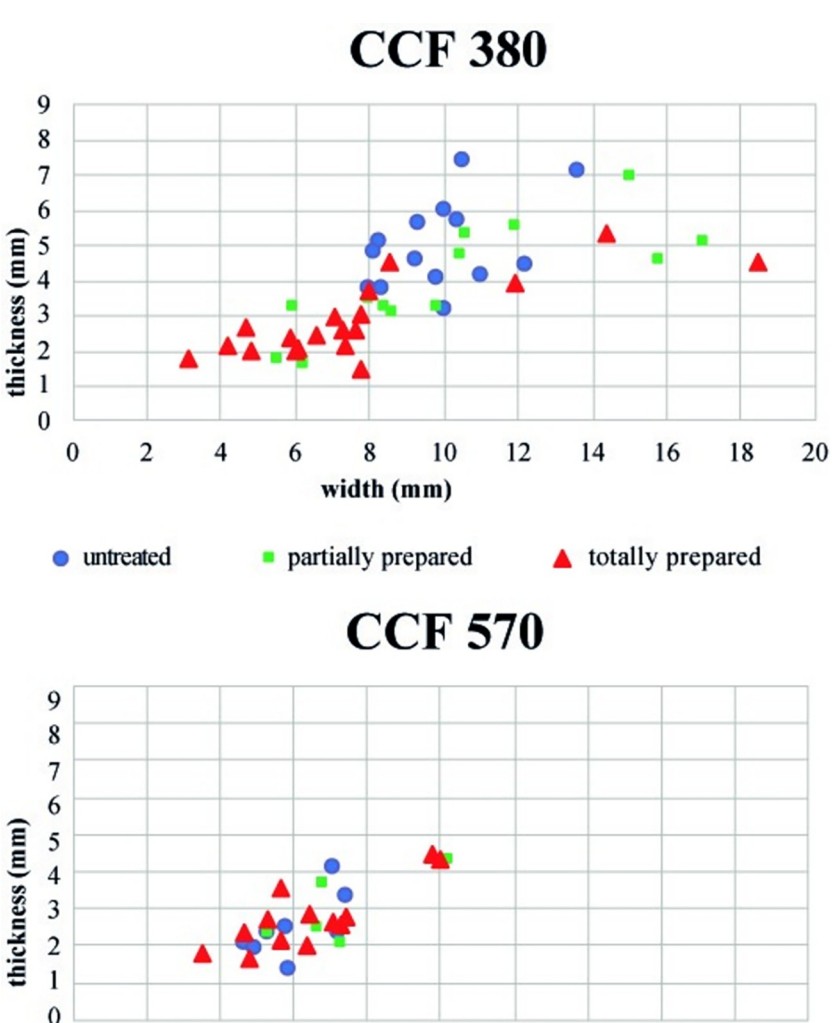

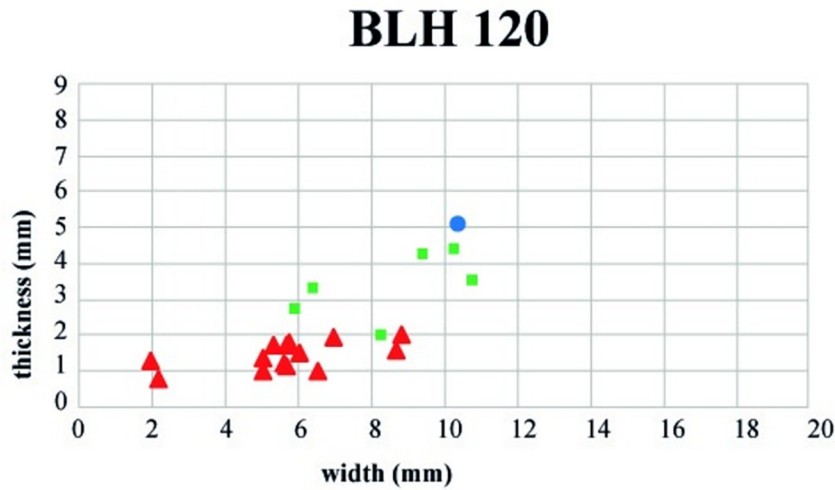

**Fig 6. Diagrams of the distribution of width and thickness of butts by type of overhang preparation per house.**

component in this House 380 at Cuiry-lès-Chaudardes. On the other hand, in House 570 at CCF, nearly all butt dimensions are concentrated within a minimal range of 2 mm, regardless of the overhang preparation (Fig 6 and Table 5). Even if a few pieces are distinct, the technical practices of this house seem to have been quite homogeneous.

Our comparison of overhang preparation procedures, butt types and butt dimensions thus contributes new information:

- two houses, BLH 120 and House 380 at Cuiry-lès-Chaudardes, contain two distinct components;

- in contrast to House 570 at CCF, which has only one component.

Therefore, among the three habitation units studied, we can distinguish five components. (Fig 7).

To advance our understanding of these distinct components within the same habitation units, we introduced two secondary blade debitage criteria: platform angle and blade detachment order (Fig 8). We should note that the samples are small, but though they do not enable statistical validity, they allow us to detect significant trends that contribute to the discussion.

In House 120 of BLH, a clear opposition appears between the two platform angle analysis components. The platform angle of most of the blades with totally prepared overhangs is close to 70˚, while the angles of the second set are nearly 90˚. This clear opposition is, nevertheless, associated with very similar detachment order proportions. There is no difference between the two blade sets. Blades with a trapezoidal section, with a slight majority of 212' orders, were produced.

In House 380, the opposite is true. The platform angles are similar for the two groups. On the other hand, the detachment orders clearly distinguish them. The group with mostly well-prepared blades has a very high proportion of 212' orders, showing that the knapper was fully capable of implementing specific debitage orders that enabled the recurrent production of regular blades with a trapezoidal section.

This analysis reinforces the opposition between the two components identified in the 120 Houses of BLH and 380 of Cuiry. The substantial distinction in the inclination of the striking platforms of the two BLH components and relative to the CCF component distinguishes the knapper of these blades with heavily prepared small butts from the rest of the production. For House 380 of Cuiry, the distinction based on the blade detachment orders also distinguishes the blades with more intensively prepared butts. The knapper responsible for this production had the knowledge and skills to obtain numerous regular blades with a trapezoidal section.

To conclude, we will discuss the qualitative criteria specifically employed to more thoroughly describe the platform edge preparations. These criteria support the distinctions detected between these five components.

House 570 of CCF is the most homogeneous of the assemblages analyzed, and the association between the butt dimensions and preparations is homogeneous, as are the detachment orders and the platform angles. House 570 is thus a coherent assemblage with no significant distinctions according to the criteria analyzed. The qualitative aspect of the detachment preparations is based on their location, the extension of the preparation and the intensity of the removal of the edge of the striking platform for the partially or totally prepared butts. In House 570, the abrasion extension is minimal, meaning that the preparation flakes are only slightly invasive in 15 out of 20 cases (Fig 9-1). We also identified three blades with overhang abrasion in the direction of the striking platform, a feature observed only in this house.

For House 380 of CCF, the situation is very different because the assemblage is not homogeneous. Component 1, consisting of blades with wide and thick butts, with untreated or

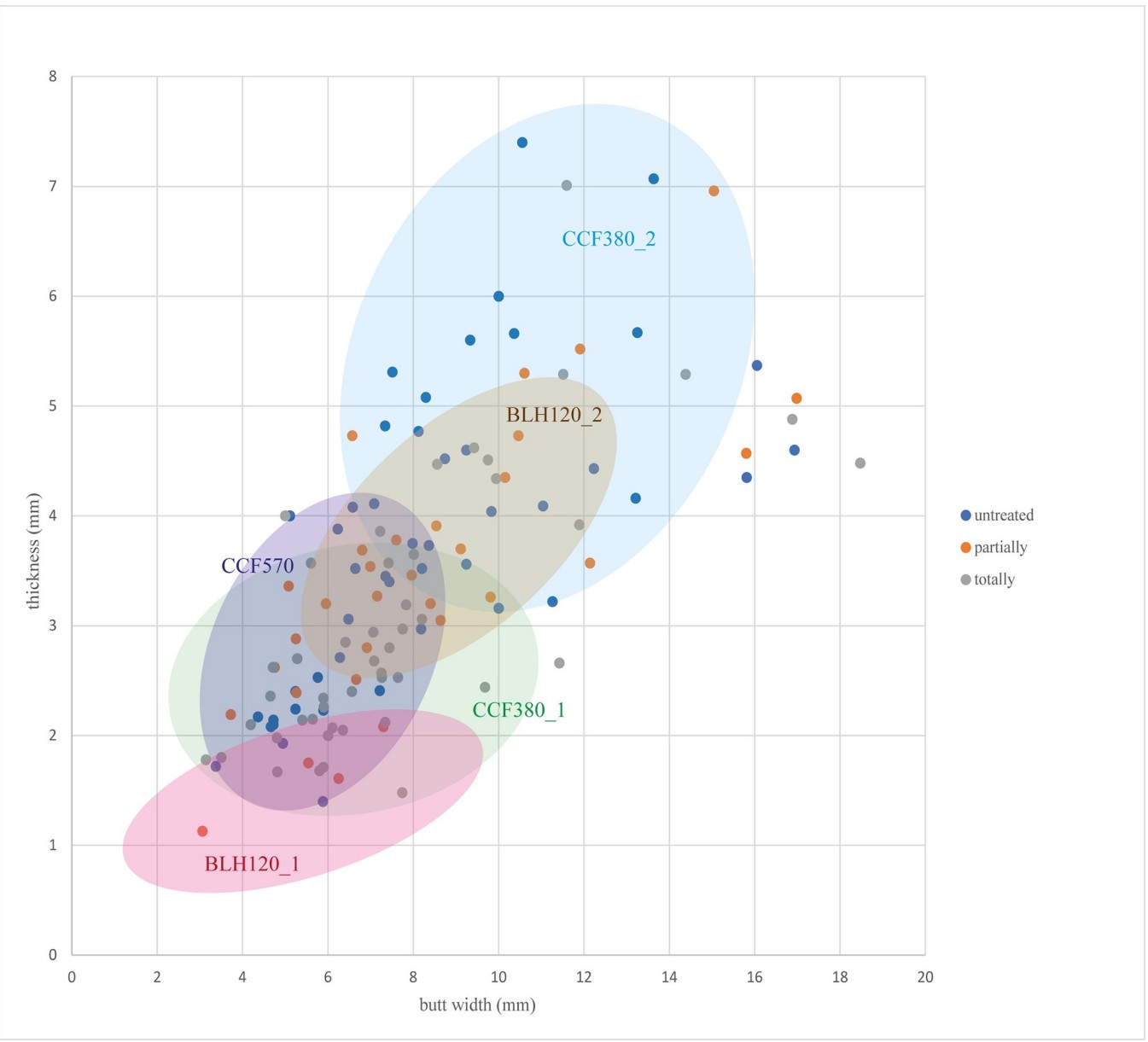

**Fig 7. Comparison of ranges of the 5 recognized groups based on butt measurements by overhang preparation type.**

partially prepared overhangs (Figs 9-2a and 10), is distinct from Component 2, which includes blades with abraded overhangs and thinner, narrower butts, and a 212' preferential detachment order. For this component, the striking platform abrasion modalities have a unique feature: the preparation is meticulous, complete, and accompanied by one or two lamellar detachments, 4 to 7 mm long (12 out of 17 blades, Fig 9-2b). We observed such elongated detachments in only 1 case out of 20 blades in H570 and 1 case out of 15 in H120 of BLH. This feature can thus be considered as typical of Component 1 of House 380.

House 120 of BLH displays this same bipartition, distinguishing a group with wide and thick butts with little or no preparation and a group with systematically abraded butts and very small dimensions. While the detachment orders are similar, the platform angle is very distinct

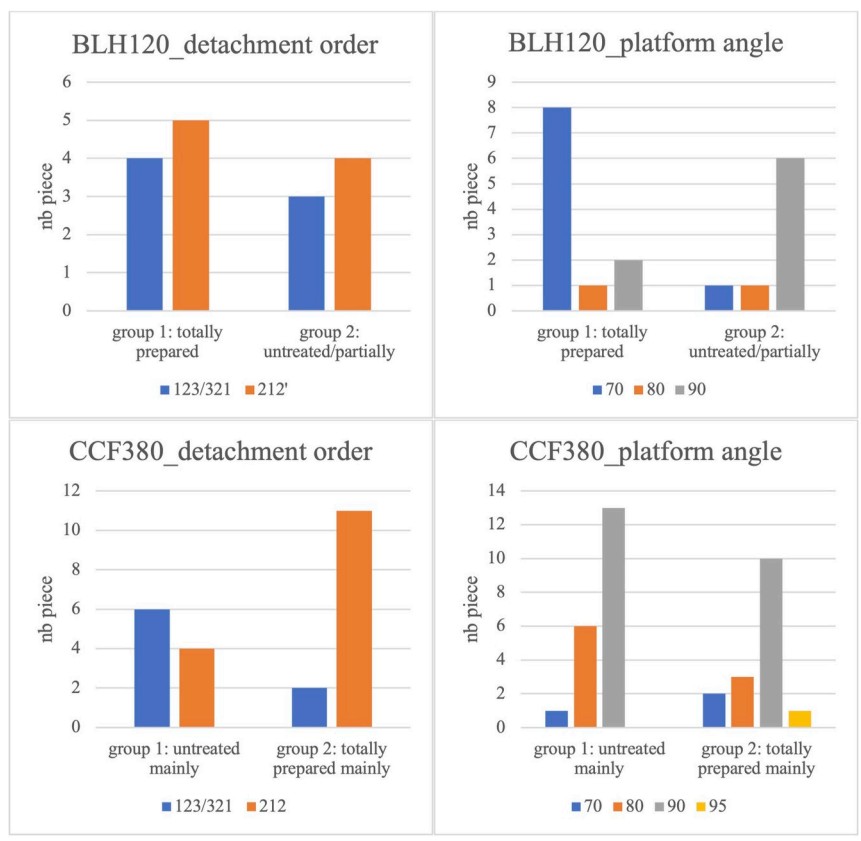

**Fig 8. Histograms of codes (detachment order) and platform angles (in degrees) according to the different groups of blades.**

in Component 1 (8 pieces out of 11 have an angle near 70˚). This is a characteristic feature of this component. In effect, only 3 of the 36 cases were observed in House 380, while no piece with an angle this small was observed in House 570. But this is not the only distinct feature in this assemblage as the striking platform edge preparation is also unique. The butts are plain and the abrasion is intensive, resulting in a significant narrowing of the striking platform and creating narrow, linear, or sometimes filiform butts (Fig 11). These features are found on blades made from various materials, including the coarsest-grained quartzitic sandstone (Fig 12).

This last observation confirms that these distinctions do not reflect different methods of treating various materials. Each of these features are distributed indifferently in each of the distinguished raw materials (Table 6).

Therefore, our detailed study of the technical characteristics of the blades in three LPC habitation units at the sites of BLH and CCF demonstrates the coexistence of five clearly distinct components.

## Discussion: Idiosyncratic manifestations in the blade debitage

### Five components reflecting the work of individual knappers

We have revealed the existence of variability in the laminar productions in the three LPC domestic units studied. Five components are clearly distinct from each other: two in House 120 of Bucy-le-Long; two in House 380 of Cuiry-lès-Chaudardes, and one in House 570 of Cuiry-lès-Chaudardes.

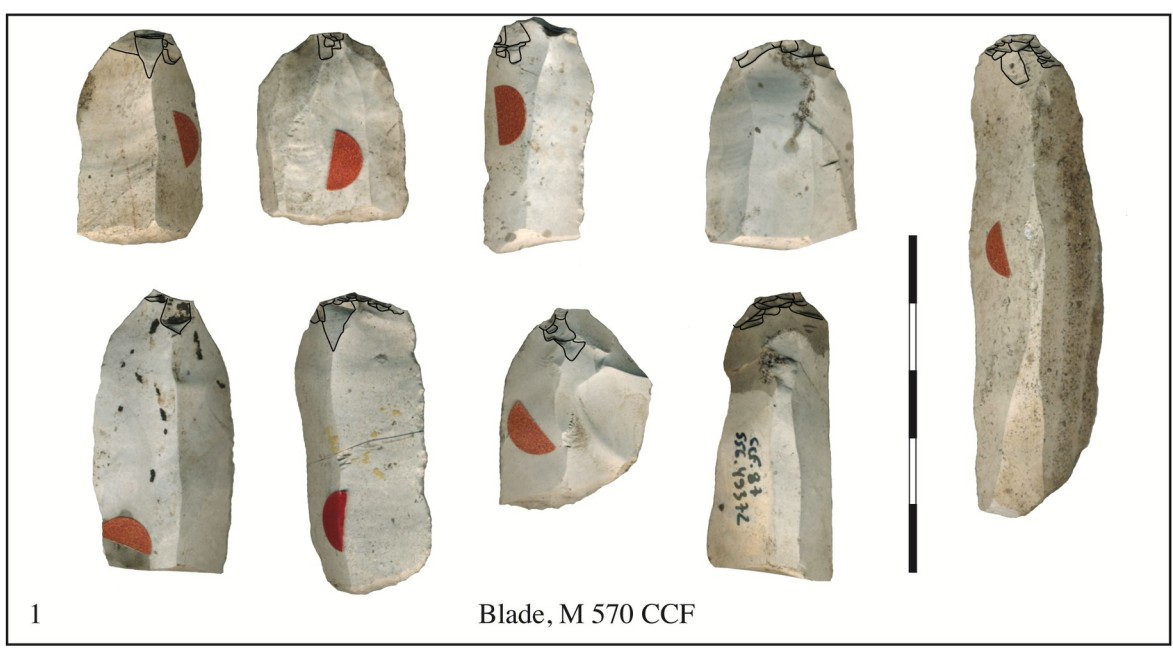

1 Blade, M 570 CCF

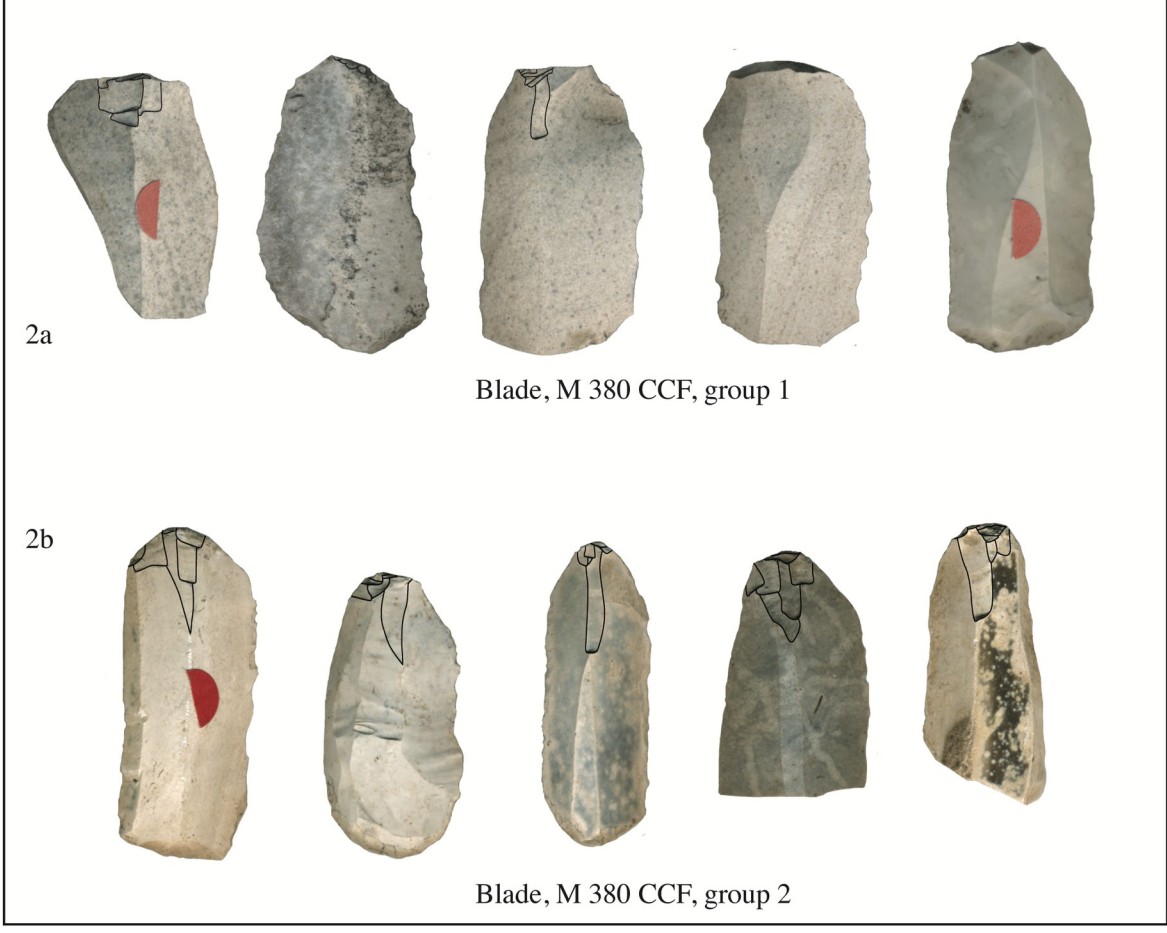

2a

Blade, M 380 CCF, group 1

2b

Blade, M 380 CCF, group 2

**Fig 9. Qualitative aspect of the detachment preparations for the blades of Cuiry-lès-Chaudadres.** In House 570, the abrasion extension is minimal, meaning that the preparation flakes are only slightly invasive (1). In House 380 of CCF, group 1, consisting of blades with wide and thick butts, with untreated or partially prepared overhangs (2a), group 2, includes blades with abraded overhangs and thinner, narrower butts, and the striking platform abrasion modalities have a unique feature: The preparation is meticulous, complete, and accompanied by one or two lamellar detachments (2b), (CCF, M 570, M 380, Soissons, France).

The two components individualized at Bucy-le-Long are clearly distinct from each other. Component 1 displays technical features that have not been identified elsewhere and do not correspond to the description proposed for the Alpha technical tradition, *a priori* dominant in the Paris Basin. The inclination between the striking platform and blade detachment surface suggests a very different core morphology with a seemingly plain striking platform.

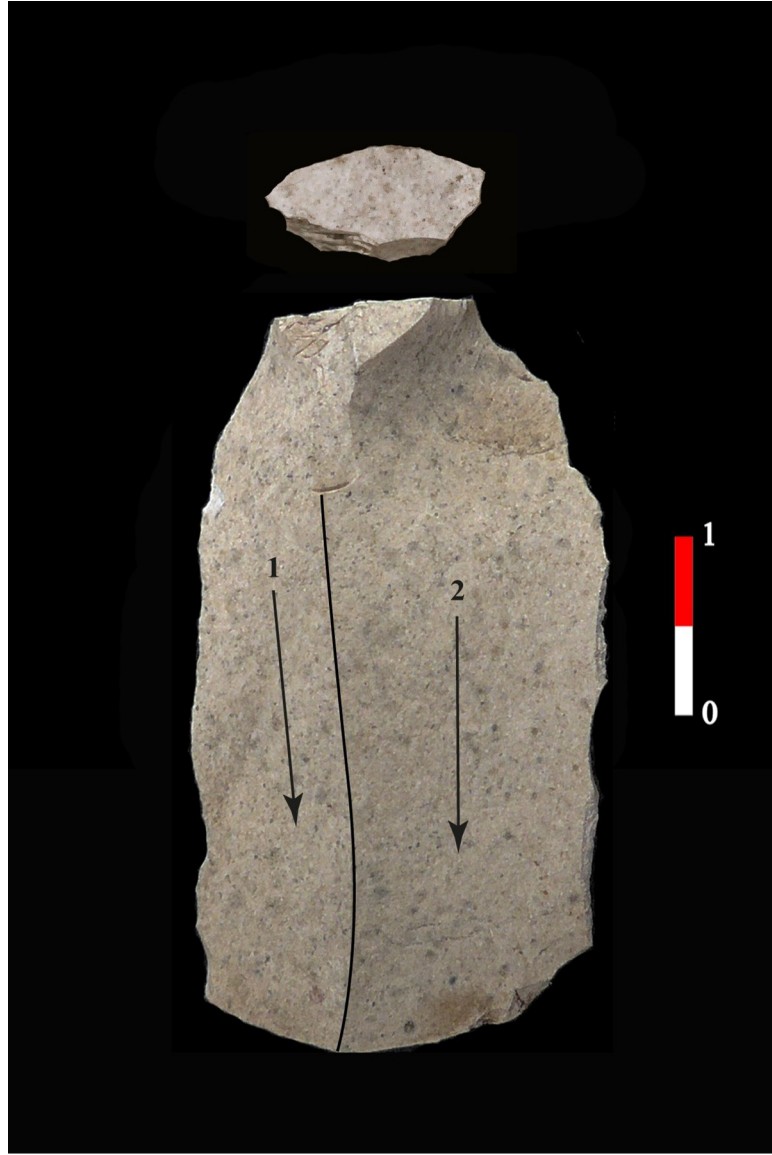

**Fig 10. Example of blade of the group 1 in the H380 of Cuiry-lès-Chaudardes, (CCF, 378–44583, Soissons, France).**

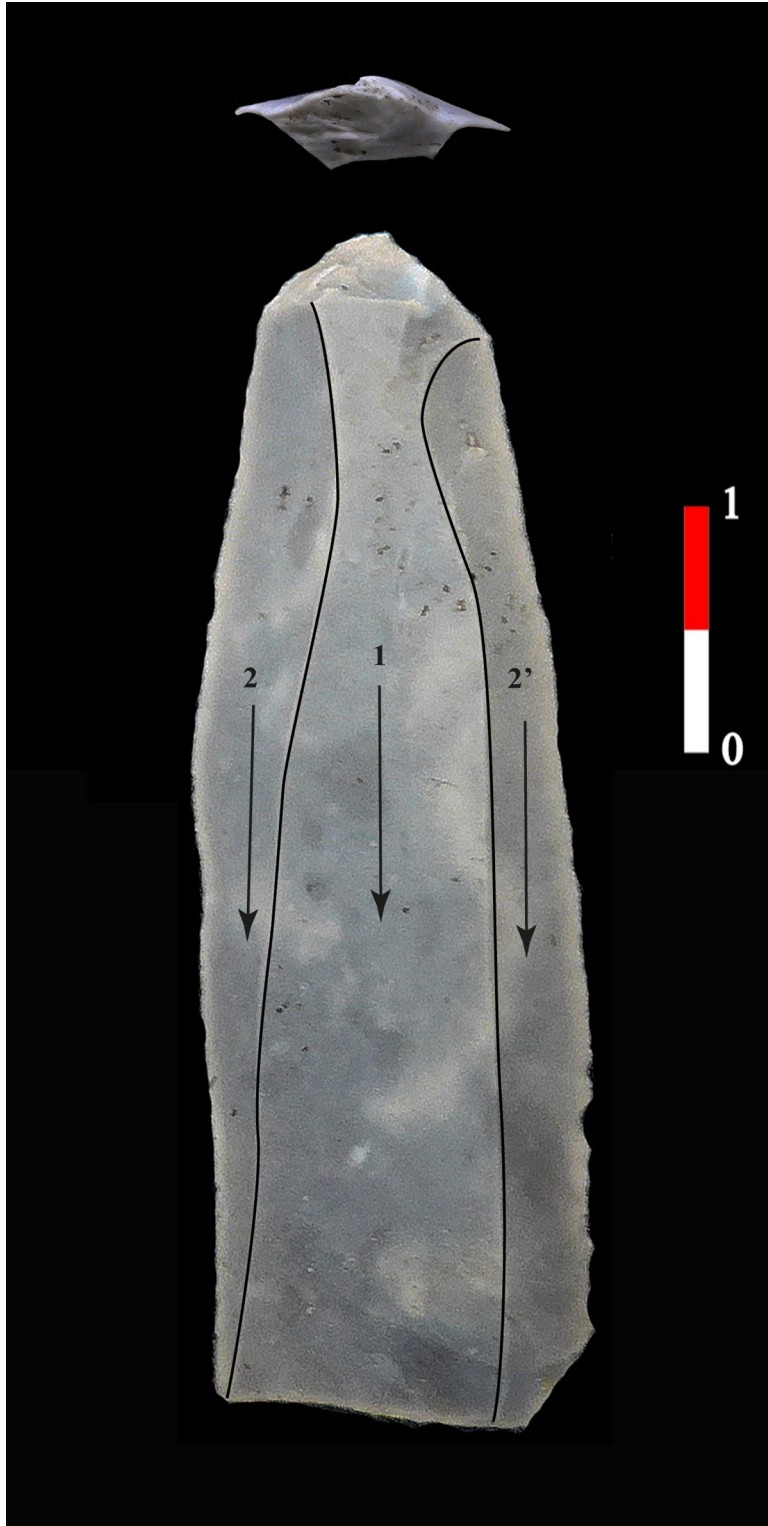

**Fig 11. Blade of Cretaceous flint in the House 120 of Bucy-le-Long "La Héronnière".** Blade of group 1 where abrasion is intensive, resulting in a significant narrowing of the striking platform and creating narrow, linear, or sometimes filiform butts, (BLH, 122–81, Soissons, France).

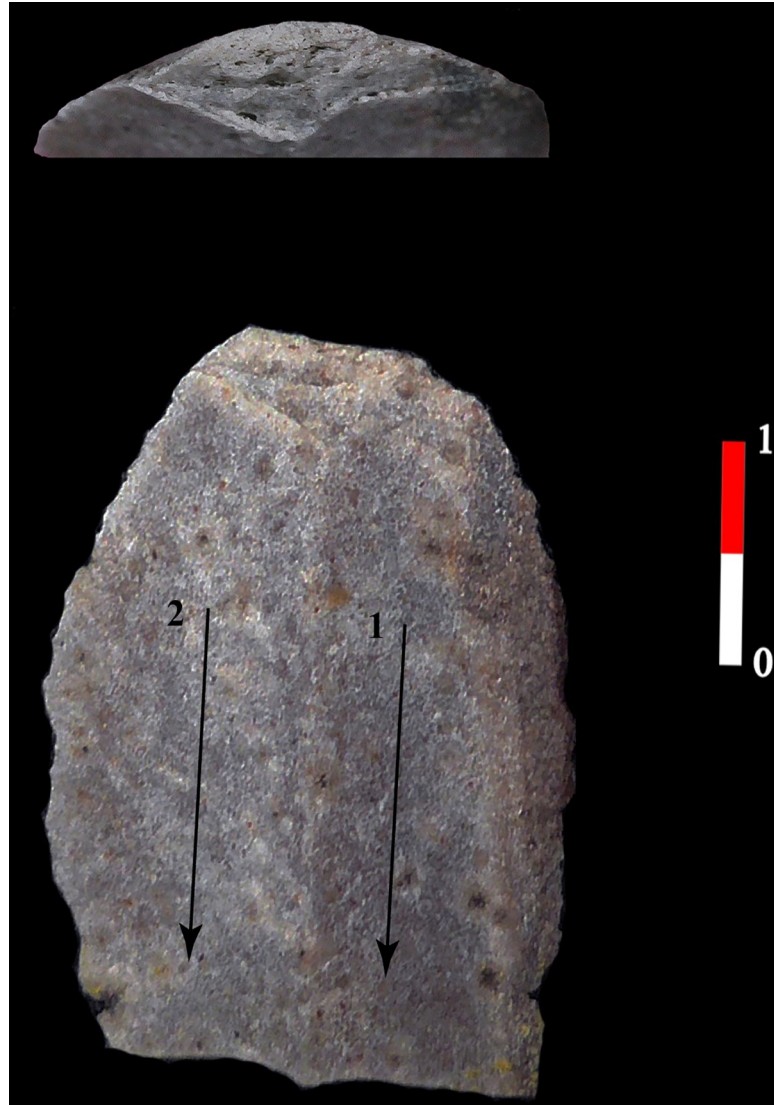

**Fig 12. Blade of quartzite in the House 120 of Bucy-le-Long "La Héronnière".** Another blade of group 1, the abrasion is also intensive, which shows that the raw material is not linked to this type of preparation, (BLH, 124–821, Soissons, France).

Furthermore, the knapper heavily abraded the overhangs and striking platform edge on this core, resulting in a massive reduction of the surface in contact with the punch as the blade was detached. Therefore, there are several unique technical features that do not correspond to the

**Table 6. Raw materials according the different groups of blades.**

|  | BLH 120 | | CCF 380 | | CCF 570 |
|---|---|---|---|---|---|
|  | group 1: totally prepared | group 2: untreated/partially | group 1: untreated mainly | group 2: totally prepared mainly | one group |
| Senonian | 9 | 4 | 3 | 9 | 17 |
| Bartonian | 3 | 3 | 16 | 9 | 7 |
| Turonian |  | 1 |  | 1 |  |
| quartzite | 2 |  | 1 |  |  |
| total | 12 | 8 | 20 | 19 | 24 |

modalities currently identified for the Alpha tradition. The absence of platform faceting and the different geometry of the core are indeed noticeable and could not be related to distinctive level of know-how or to the production of specific tools. Furthermore, these gestures are not belonging to the technical repertoire of knappers from Alpha tradition. This indicates that there could sign a new technical tradition to be further explored in the future, or a form of local invention. We will indeed have to track these technical gestures in the near future. If we can track them on another site, it would sign technical practices from another learning network. If not, we will have to reconsider this hypothesis to discuss a possible form of local variability.

The second component (BLH 120_2) has features quite similar to those of House 380 of Cuiry (CCF380_1): more or less light preparations and a large range of butt dimensions. The technical features correspond perfectly with those of the Alpha technical tradition.

Therefore, the duality observed in this habitation unit (BLH 120) corresponds to two individuals who probably learned to work flint in different communities of practice.

House 380 of CCF all displays two components (Figs 7 and 9). The first one (CCF 380_2) is represented by partial or absent preparation and a punch placement well behind the striking platform. The butts are thus wide and thick, often with an untreated external overhang and a dominant 123/321 blade detachment order. Here, the technical features are consistent with the Alpha technical tradition. The second component (CCF 380_1) is distinct due to its careful preparations and narrow butt. In addition, in 15 out of 20 cases, the overhang abrasion is accompanied by elongated lamellar detachments. As shown by the experimenters, this preparation is not trivial as there is a link between careful preparation, which reduces the dimensions of the butt and thus concentrates the energy of the percussion, and a greater regularity in the laminar products. A strict 212' detachment order accompanies this meticulous preparation, showing perfect knowledge and mastery of specific arrangements [75,76] that enable a recurring production of regular blades with a trapezoidal section. These technical features also correspond to the Alpha tradition. The variability exists in:

- the butt dimensions, correlated with the care taken in their preparation;

- the mastery or not of specific debitage arrangements, and;

- the presence of a specific action or "obsession" according to [24] in Component 1.

The two first criteria are closely linked to the skill level of the knappers. Indeed, high level of skills can be perceived through "very low metric variation in artifact size" and "symmetric cross-section" [77]. The specific arrangements are not only difficult to master and implement but 212' codes are related to blades with regular trapezoidal cross-sections [75]. This criterion is relevant to identify high level of skill [77]. The second element highlighted here is butts' dimensions. Indeed, the more the butt is small, the more the blade is regular. This regularity reflects the "very low metric variation size", qualifying productions of highly skilled knappers. Also, in this house, the two components seem to reflect two different skill levels within a single technical repertoire (Alpha tradition). We described above the discovery context in which the assemblages seem to have resulted from a rapid filling of the pits. This situation contradicts the attribution of these two components to a single individual with increasing skill through time. This is also supported by the identification of a specific action (or "obsession" [24]), which we find in only one of the two components.

Finally, in House 570 of CCF, this study reveals laminar products within a very coherent assemblage (Fig 7), regardless of the preparation type, keeping in mind that most of the butts have a totally abraded overhang with a preparation that is only slightly invasive. The regularity of the impact zone dimensions, the small range of variability of the detachment orders, and striking platform angles according to preparation type suggest that they were the work of a

single knapper. This argument is supported by the fact that this group of blades has a much smaller dimensional interval than the other houses (Fig 6). This indicates, moreover, the high skill level of the knapper. The technical features are thus in accordance with the definition of the Alpha technical tradition. In this case, the variability of the striking platform edge interventions is linked to the correct placement of the punch, case by case, during the evolution of the debitage procedure. There is no systematic abrasion.

Therefore, we propose that the five identified components reflect the work of five different knappers. This result is even more interesting considering that the two CCF houses are contemporary, thus showing variability independent of chronology. During this same pottery stage, at least three knappers would thus have practiced at CCF. Furthermore, we can identify two house compositions that housed two knappers:

- the composition of House 380 appears to correspond to two cohabiting knappers with different skill levels, raising the question of a duality linked to the ages of the individuals;

- in contrast, House 120 of BLH seems to have been inhabited by two knappers that probably originated from two different technical traditions, one of which was not part of the Alpha tradition, which is the most frequent in the Aisne Valley LPC (Fig 13).

This confrontation between two levels of interpretation also shows that the individual variability with a single technical tradition is expressed:

- in small, specific actions, or "quirks," associated with the overhang preparations, more precisely visible for the knappers with higher skill levels (CCF 570 and CCF 380_1);

- in the skill level variability perceptible at the house scale, and;

- in the differential intensity of some actions.

It is, of course, the work performed in houses that enables us to distinguish different groups (essentially based on the butt types/degree of preparation of the detachment/butt dimensions) between habitation units with a single technical tradition repertory.

### Implications for a renewal of our interpretative perspectives concerning the organization of activities in the first agro-pastoral villages

This unique study seeking to identify distinct artisans in the villages of the first LPC agro-pastoralists is highly stimulating for future research. Only the small size of the samples will hinder the potential for a renewal of our interpretative perspectives on the organization of activities in the villages.

We have shown that the production of House 570 was realized by one knapper, in contrast to Houses 380 of CCF and 120 of BLH, for which we identified two knappers. This result is even more relevant because these two buildings are large houses, unlike House 570, confirming, firstly, the identification of a domestic production organization during this period [2,64] and secondly, that the large buildings accommodated a larger number of persons/extended family, in contrast to the smaller houses. The data on the flint industries and, more precisely, on the social distribution of activities will thus contribute to the models already proposed for the site of Cuiry-lès-Chaudardes (e.g., [78]).

## Conclusion

The method we propose here addresses the modalities employed in the preparation of blades for detachment. We thus focused on analyzing the striking platform edge preparation and the

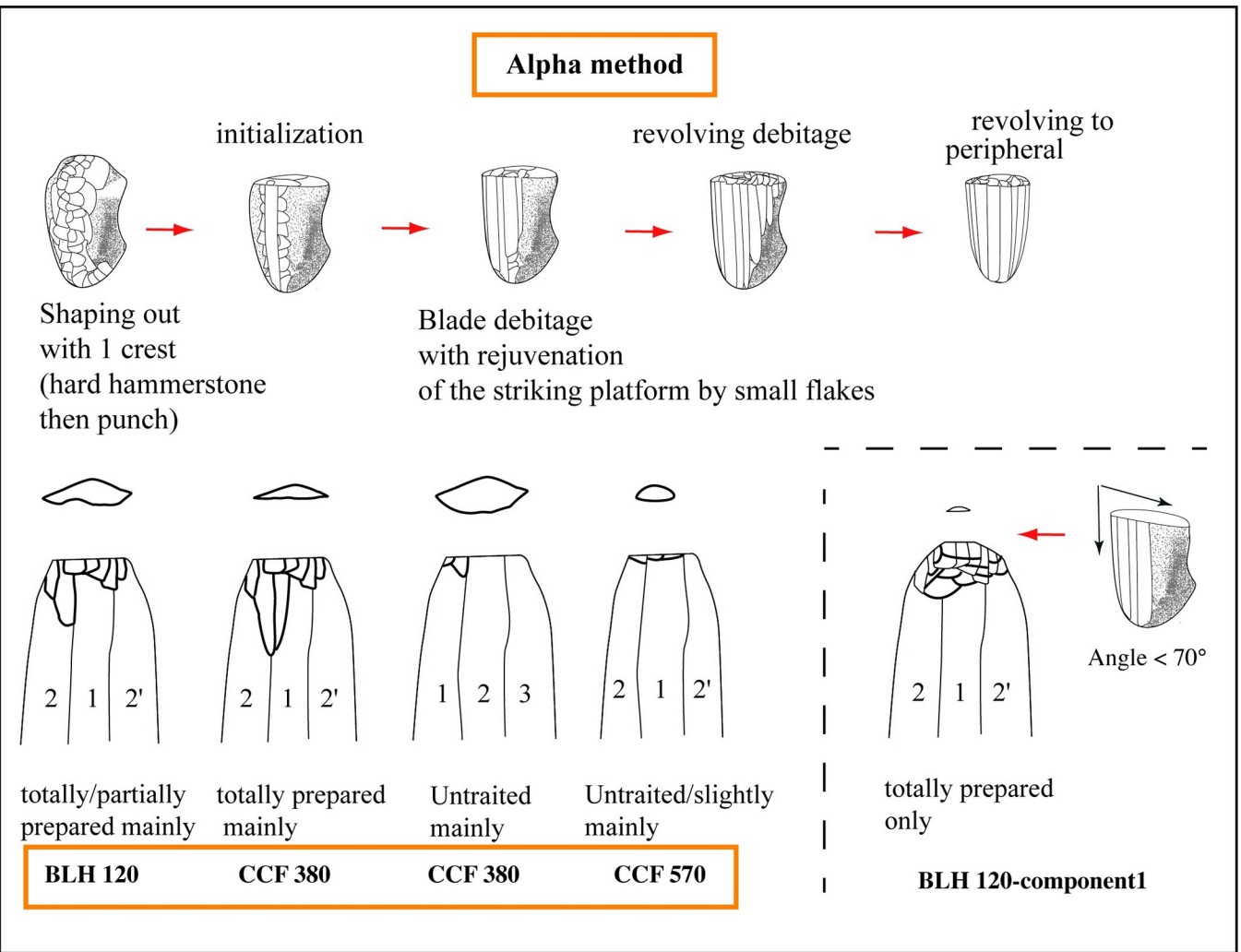

**Fig 13. Schematic illustration of the Alpha knapping method and the individual variability of the knappers.** This figure shows the variability of preparation methods for the the flint blades. Nevertheless, these differences belong to the modalities currently identified for the Alpha tradition. Into the house component 1 of BLH, the inclination between the striking platform and blade detachment surface suggests a very different core morphology with a seemingly plain striking platform, probably not belonging to the Alpha tradition.

morphologies of the impact zone. Previous studies have already shown that analyses of the proximal parts of laminar products can contribute significant information [12,54,58–60].

In our study, this method and other criteria enabled us to identify different technical traditions in the Early Danubian Neolithic in the Rhine/Meuse/Seine basins.

Our study of three habitation units in two sites in the Aisne Valley shows that it is possible to distinguish different assemblages, which we interpreted the work of different knappers. This study documents the organization of flint knapping in the LPC houses of Bucy-le-Long "La Héronnière" and Cuiry-lès-Chaudardes "Les Fontinettes." Such idiosyncratic manifestations have already been observed in other contexts, mainly Upper Paleolithic ones, such as the Magdalenian at the sites of Etiolles and Pincevent [24,26]. These studies also included the notion of varying skill levels in executing actions [27]. We demonstrated that this notion could be fundamental to distinguishing individuals belonging to a single community of practice.

At the scale of our study region and for the period considered, we are not aware of any similar studies. Other studies have focused, however, on demonstrating the existence of flint

knapping specialists in some villages of the Blicquy/Villeneuve-Saint-Germain group [46,79]. These studies are based on a technological approach that broadly distinguishes, in some villages, two types of laminar productions, one considered as domestic and the other as the work of a specialist and dedicated to exchanges between villages. The systematic approach that we applied here should support this distinction of productions per house. For the LPC, the autonomy of the relationships between houses has been addressed, especially in research on the Rhine/Meuse region [1,7,80]. There are probably several scenarios in western Europe, with the house displaying the most evidence for different productions (except for exogenous products, of course). However, the statistical approach applied to the site of Elsloo in the Netherlands suggests that during some occupation phases of this site, certain houses took charge of the flintknapping activities and redistributed the blades [7]. We do not know, however, if these activities involved several knappers or not. Our approach can significantly contribute to these research questions.

Finally, the models of the socio-economic organization of activities developed for the village of Cuiry-lès-Chaudardes have demonstrated an opposition between large and small domestic units [78,81]. Future studies in the framework of the Homes ANR (direction C. Hamon, UMR 8215) will contribute to a clearer understanding of the organization of the producers and the identification of the actors.

We believe that detailed technological analysis seeking to identify individuals can open new avenue to discuss socio-economic organization of the first farmers. Indeed, it would be possible to formulate hypothesis about (i) the composition of the household based on the number of producers; (ii) the social origin of the household producers (same or different learning network). Integrated studies on Western LBK sites have already suggested socio-economic models proposing the opposition between long and small houses [78,81]. House size reflects varying degrees of economic maturity and particular functional status [78]. Lithic production was not included in this modelling. But our work seems to support this opposition between long and small houses. The mastering of blade production seems to involve more individuals in long houses. Furthermore, we highlighted two models of coexistence within the longhouses where knappers can be originated from the same learning networks or not. So, we have demonstrated here the potential of future integrated research which will be done in the framework of the ANR HOMES Research Project.

## Supporting information

**S1 File.**
(PDF)

## Author Contributions

**Investigation:** Pierre Allard.

**Methodology:** Pierre Allard, Solène Denis.

**Writing – original draft:** Pierre Allard, Solène Denis.

**Writing – review & editing:** Pierre Allard, Solène Denis.

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
