## [Decision Letter · Decision Letter 0]

26 Jan 2022

PONE-D-21-37152Technical traditions and individual variability in the Early Neolithic: Linear Pottery Culture flint knappers in the Aisne Valley (France)PLOS ONE

Dear Dr. ALLARD,

Thank you for submitting your manuscript to PLOS ONE. After careful consideration, we feel that it has merit but does not fully meet PLOS ONE’s publication criteria as it currently stands. Therefore, we invite you to submit a revised version of the manuscript that addresses the points raised during the review process. Both reviewers are positive, but each of them makes suggestions for additions that will substantially improve the paper. Please address all of the reviewers suggestions while making your revisions.

We look forward to receiving your revised manuscript.

Kind regards,

John P. Hart, Ph.D.

Academic Editor

PLOS ONE

Journal Requirements:

2. In your manuscript, please provide additional information regarding the specimens used in your study. Ensure that you have reported specimen numbers and complete repository information, including museum name and geographic location. 

For more information on PLOS ONE's requirements for paleontology and archaeology research, see https://journals.plos.org/plosone/s/submission-guidelines#loc-paleontology-and-archaeology-research.

(NO, The funders had no role in study design, data collection and analysis, decision to publish, or preparation of the manuscript)

4. We note that Figure 1 in your submission contain map images which may be copyrighted. All PLOS content is published under the Creative Commons Attribution License (CC BY 4.0), which means that the manuscript, images, and Supporting Information files will be freely available online, and any third party is permitted to access, download, copy, distribute, and use these materials in any way, even commercially, with proper attribution. For these reasons, we cannot publish previously copyrighted maps or satellite images created using proprietary data, such as Google software (Google Maps, Street View, and Earth). For more information, see our copyright guidelines: http://journals.plos.org/plosone/s/licenses-and-copyright.

Reviewers' comments:

Reviewer's Responses to Questions

**Comments to the Author**

1. Is the manuscript technically sound, and do the data support the conclusions?

Reviewer #1: Yes

Reviewer #2: Yes

2. Has the statistical analysis been performed appropriately and rigorously? 

Reviewer #1: Yes

Reviewer #2: Yes

3. Have the authors made all data underlying the findings in their manuscript fully available?

Reviewer #1: Yes

Reviewer #2: Yes

4. Is the manuscript presented in an intelligible fashion and written in standard English?

Reviewer #1: Yes

Reviewer #2: Yes

5. Review Comments to the Author

Reviewer #1: This contribution is an exciting contribution for the study of particular lithic productions (blade technologies), in which the identification of potential differences between the assemblages is pursuit. The paper's general aim and primary objective are interesting and important since most of the works do not pay attention to socio-cultural aspects related to particular methods and variabilities applied in the flaking. The paper demonstrates the high preparation and knowledge of the authors in the technical reading of these materials. However, this circumstance omitted some crucial aspects in the contribution.

Some of the aspects evaluated as discriminant markers in the material include as mention by the authors: "The inclination between the striking platform and blade detachment surface suggests a very different core morphology with a seemingly plain striking platform. However, the knapper heavily abraded the overhangs and striking platform edge on this core, resulting in a massive reduction of the surface in contact with the punch as the blade was detached." Those differences are understood as "different technical traditions" and not just learning skills or simply individual expressions in the knapping. I have the same opinion about the existence of such differences, but at the same time, I thought that such variations could happen inside the same knapping school. If the authors disagree with this, it is crucial to justify their interpretation in a deeper way.

Another aspect of being considered is the chronology of the technical evolution of particular markers and knapping resources that may change in short periods. From the moment that the study includes discrete clusters, it is essential to confirm the coetaneous character of the studied assemblages. Regarding this topic, please improve the chronological description of the studied units and their contemporaneity.

I agree with the authors in the general appreciation of the value of technical resources as an informative element in comparison to different assemblages or lithic cluster, but traditions have to be considered from a whole perspective (that include the change in the technical resources employed along the entire C.O.) and not just a particular preparation mode.

The existence of skill levels is slightly presented, and the criteria used are not clear enough (lines 583 to 589). Try to clarify the variables employed and include references about the previously employed criteria.

Please include in the discussion the potential effect of changes in the raw material quality in the difference in the technical strategies documented (raw material discussion and description are mostly absent in the text).

There is a general absence of the review of experimental blade production in this chrono-cultural context. Please, include a more intense review of previous experimental studies and their conclusions (check the following references based on previous experimental and skill analysis:

-Castañeda, N. A 2016. Few Good Blades. An Experimental Test on the Productivity of Blade Cores from the Casa Montero Early Neolithic Flint Mine (Madrid, Spain). Journal of Lithic Studies 3(2).

-Soriano S., 2019. Un néandertalien qui avait plus d’un coup de percuteur dans son sac ! Le procédé de l’éclat-punch. In: David E (ed) Anthropologie des techniques. Cahier 1 : De la mémoire aux gestes en préhistoire. L’Harmattan, Paris, pp 157–166.

- Baena, J. et al. 2019. Good and Bad Knappers Among Neanderthals Pages 95-117

In Learning Among Neanderthals and Palaeolithic Modern Humans, Yoshihiro Nishiaki &Olaf Jöris Eds, 2019 Springer.

In any case, the study is quite intelligent and clearly opens a precise methodology for approaching different sin styles, probably in past knapping schools and perhaps in cultural traditions. With this contribution, the authors start the discussion about these aspects, and it is quite worthy of initiating the debate.

I recommend its publication after some minor changes since the perspective and the estudy is woth enough.

As minor changes I would suggest the following aspects:

1) Please include changes in figures 2,3 and 4 and also 10, 11, and 12 with images of the complete blades

2) Including the directions of the detaches of preparations (and perhaps the diachritic) in the previous figures will also provide a clearer image of the technical differences.

3) Study the possibility to integrate a map of the different traditions described in the text with chronologies and locations.

4) Please consider to present a more clear way the briefly described “Alpha tradition.” And the dif

Reviewer #2: The article is one of the few original studies that is focused on the search for meaningful criteria that make it possible to characterize not only the general and the specific in the methods of production of flint blades - blanks for tools - at various sites of the Vaux-et-Borset and the Aisne Valley regions, but to determine individual techniques in flintknapping inherent in particular masters. The author describes in sufficient detail the methodological basis of the research, based on a rigorous analysis of flint blades: their parameters, types of striking platform, the angle between the platform and the flaking surface. Involving the results of experimental work, he reveals the details of the flint splitting process and naturally comes to the conclusion that the artifacts he studied with the same flint splitting technology (pressure or impact) have significant differences, allowing them to be considered products of different masters. The results obtained allowed the author to make a logical conclusion that the technological analysis of the flint inventory is an important source not only for highlighting the history of technology, but it also allows us to consider complex issues of socio-economic features of the life of ancient societies.

All of the above allows to positively evaluate the peer-reviewed article. Nevertheless, the addition of some sections will significantly increase the argumentation of the author's proposals and reveal to the reader the significance of the work done.

In “Materials and Methods”:

it is necessary to give a short description of the flint industry of that local variant of the LPC, to which the settlement materials belong; give a more detailed description of the archaeological context, as well as indicate what other categories of flint products and in what quantity, besides the blades studied by the author, were found in residential and household objects (cores, retouchers, tools used for flintknapping?);

In “Results”

it is necessary to more reasonably justify the assumption that the difference in the methods of obtaining blades found in the same house may indicate the work of two different masters. Modern experimental work suggests that the same master can successfully modify some operations to obtain the same type of flint blank for various objective (quality of raw materials, condition and quality of a hammer stone, etc.) and subjective reasons (the need to speed up the process, fatigue, etc. ).

The “conclusion” section could be expanded by proving the importance of a detailed technological analysis of the flint inventory for modeling various socio-economic reconstructions.

The conclusions can be substantially supplemented by the inclusion of schematically depicted flint processing techniques that characterize the individual details of flint processing.

I would like to wish the author to continue the work begun, to increase the number of materials studied, which is of great importance for the study of various aspects of production activities.

6. PLOS authors have the option to publish the peer review history of their article (what does this mean?). If published, this will include your full peer review and any attached files.

Reviewer #1: **Yes: **Concepcion Torres Navas

Reviewer #2: No

---

## [Author Response · Author response to Decision Letter 0]

18 Mar 2022

Cover Letter 10th March 2022

Dear editor, dear reviewers

We are sending the revised manuscript entitled “Technical traditions and individual variability in the Early Neolithic: Linear Pottery Culture flint knappers in the Aisne Valley (France)”. 

We thank Plosone for their interest in our paper and the reviewers for their comments to improve the manuscript and the figures

We therefore respond point by point to the various remarks and suggestions of the reviewers

PLOS ONE

Journal Requirements:

2. In your manuscript, please provide additional information regarding the specimens used in your study. Ensure that you have reported specimen numbers and complete repository information, including museum name and geographic location. 

NO, The funders had no role in study design, data collection and analysis, decision to publish, or preparation of the manuscript

This study has not received any particular funding, we are collaborators in the ANR Homes program and Solène Denis is a post-doc at the University of Masaryk in the Czech Republic.

However, we have reported the writing of this article and the University of Masaryk will support us financially for the publication of the article in the journal Plosone, provided that the publication is accepted of course.

b) State what role the funders took in the study. If the funders had no role in your study, please state: 

- No salary or grant

d) If you did not receive any funding for this study, please state: 

- The authors received no specific funding for this work.

4. We note that Figure 1 in your submission contain map images which may be copyrighted

- This map has been designed with a Digital Elevation Model BD ALTI® from the French National Institute of Geographic and Forest Information (IGN) which are public data, free and open. So, there is no issue to use this map. We added the reference to the Digital Elevation Model in the caption of the figure n° 1 and 4. 

5. Please review your reference list to ensure that it is complete and correct. Any changes to the reference list should be mentioned in the rebuttal letter that accompanies your revised manuscript. 

- We have responded to the reviewers' remarks by adding references, we have added 20 references.

5. Review Comments to the Author

Reviewer #1: This contribution is an exciting contribution for the study of particular lithic productions (blade technologies), in which the identification of potential differences between the assemblages is pursuit. The paper's general aim and primary objective are interesting and important since most of the works do not pay attention to socio-cultural aspects related to particular methods and variabilities applied in the flaking. The paper demonstrates the high preparation and knowledge of the authors in the technical reading of these materials. However, this circumstance omitted some crucial aspects in the contribution.

Some of the aspects evaluated as discriminant markers in the material include as mention by the authors: "The inclination between the striking platform and blade detachment surface suggests a very different core morphology with a seemingly plain striking platform. However, the knapper heavily abraded the overhangs and striking platform edge on this core, resulting in a massive reduction of the surface in contact with the punch as the blade was detached." Those differences are understood as "different technical traditions" and not just learning skills or simply individual expressions in the knapping.

- I have the same opinion about the existence of such differences, but at the same time, I thought that such variations could happen inside the same knapping school. If the authors disagree with this, it is crucial to justify their interpretation in a deeper way.

- There is a general absence of the review of experimental blade production in this chrono-cultural context. Please, include a more intense review of previous experimental studies and their conclusions (check the following references based on previous experimental and skill analysis:

-Castañeda, N. A 2016. Few Good Blades. An Experimental Test on the Productivity of Blade Cores from the Casa Montero Early Neolithic Flint Mine (Madrid, Spain). Journal of Lithic Studies 3(2).

-Soriano S., 2019. Un néandertalien qui avait plus d’un coup de percuteur dans son sac ! Le procédé de l’éclat-punch. In: David E (ed) Anthropologie des techniques. Cahier 1 : De la mémoire aux gestes en préhistoire. L’Harmattan, Paris, pp 157–166.

- Baena, J. et al. 2019. Good and Bad Knappers Among Neanderthals Pages 95-117

In Learning Among Neanderthals and Palaeolithic Modern Humans, Yoshihiro Nishiaki &Olaf Jöris Eds, 2019 Springer.

- We group together the response to these two remarks. We have added a consequent synthesis about modern experiments, which show that the signatures of the knappers are visible (lines 122-162 of the Manuscript with Tracks). The references mentioned have been added.

Another aspect of being considered is the chronology of the technical evolution of particular markers and knapping resources that may change in short periods. From the moment that the study includes discrete clusters, it is essential to confirm the coetaneous character of the studied assemblages. Regarding this topic, please improve the chronological description of the studied units and their contemporaneity.

- The chronological seriation of the LPC sites of the Aisne valley, and more generally of the period, is mainly based on the evolution of the decorated ceramics. We have added a paragraph with references on the chronology of the studied assemblages, lines 235-241

I agree with the authors in the general appreciation of the value of technical resources as an informative element in comparison to different assemblages or lithic cluster, but traditions have to be considered from a whole perspective (that include the change in the technical resources employed along the entire C.O.) and not just a particular preparation mode.

The existence of skill levels is slightly presented, and the criteria used are not clear enough (lines 583 to 589). Try to clarify the variables employed and include references about the previously employed criteria.

- We have clarified the answer in lines 737 to 743 by adding experimental references to levels of know-how.

Please include in the discussion the potential effect of changes in the raw material quality in the difference in the technical strategies documented (raw material discussion and description are mostly absent in the text).

- We have added a paragraph on raw materials. The Paris basin is at the heart of a region rich in good quality flint, lines 474-495

In any case, the study is quite intelligent and clearly opens a precise methodology for approaching different sin styles, probably in past knapping schools and perhaps in cultural traditions. With this contribution, the authors start the discussion about these aspects, and it is quite worthy of initiating the debate.

I recommend its publication after some minor changes since the perspective and the estudy is woth enough.

As minor changes I would suggest the following aspects:

1) Please include changes in figures 2,3 and 4 and also 10, 11, and 12 with images of the complete blades

2) Including the directions of the detaches of preparations (and perhaps the diachritic) in the previous figures will also provide a clearer image of the technical differences.

3) Study the possibility to integrate a map of the different traditions described in the text with chronologies and locations.

4) Please consider to present a more clear way the briefly described “Alpha tradition.” 

- Alpha tradition is developed in lines 450-468 and we made all changes in the figures 2,3,10,11,12 with complete blades and diachritic shems. We have added also a map, Figure 4.

Reviewer #2: The article is one of the few original studies that is focused on the search for meaningful criteria that make it possible to characterize not only the general and the specific in the methods of production of flint blades - blanks for tools - at various sites of the Vaux-et-Borset and the Aisne Valley regions, but to determine individual techniques in flintknapping inherent in particular masters. The author describes in sufficient detail the methodological basis of the research, based on a rigorous analysis of flint blades: their parameters, types of striking platform, the angle between the platform and the flaking surface. Involving the results of experimental work, he reveals the details of the flint splitting process and naturally comes to the conclusion that the artifacts he studied with the same flint splitting technology (pressure or impact) have significant differences, allowing them to be considered products of different masters. The results obtained allowed the author to make a logical conclusion that the technological analysis of the flint inventory is an important source not only for highlighting the history of technology, but it also allows us to consider complex issues of socio-economic features of the life of ancient societies.

All of the above allows to positively evaluate the peer-reviewed article. Nevertheless, the addition of some sections will significantly increase the argumentation of the author's proposals and reveal to the reader the significance of the work done.

In “Materials and Methods”:

it is necessary to give a short description of the flint industry of that local variant of the LPC, to which the settlement materials belong; give a more detailed description of the archaeological context, as well as indicate what other categories of flint products and in what quantity, besides the blades studied by the author, were found in residential and household objects (cores, retouchers, tools used for flintknapping?);

We have given a more precise description of the context of the excavations in the Aisne valley and the material found in these houses lines : 196-218

In “Results”

it is necessary to more reasonably justify the assumption that the difference in the methods of obtaining blades found in the same house may indicate the work of two different masters. Modern experimental work suggests that the same master can successfully modify some operations to obtain the same type of flint blank for various objective (quality of raw materials, condition and quality of a hammer stone, etc.) and subjective reasons (the need to speed up the process, fatigue, etc. ).

- It is perfectly true that knappers can make variations, but they are always within a range controlled by their expertise. We have given several examples of studies of archaeological series which show, on the contrary, that the variations are always small. We group together the response to the two remarks of the first reviewer. We have added a consequent synthesis about modern experiments, which show that the signatures of the knappers are visible (lines 122-162 of the Manuscript with Tracks).

The “conclusion” section could be expanded by proving the importance of a detailed technological analysis of the flint inventory for modeling various socio-economic reconstructions.

The conclusions can be substantially supplemented by the inclusion of schematically depicted flint processing techniques that characterize the individual details of flint processing.

I would like to wish the author to continue the work begun, to increase the number of materials studied, which is of great importance for the study of various aspects of production activities.

- We have completed the conclusion and added a new schematic figure to clarify our point (Figure 13).

We would like to thank once again the editor and the two reviewers for their comments and support for our study. We hope to have responded to their various remarks by making a number of additions to the first version of this article.

Best regards

Pierre Allard et Solène Denis

---

## [Decision Letter · Decision Letter 1]

1 May 2022

Technical traditions and individual variability in the Early Neolithic: Linear Pottery Culture flint knappers in the Aisne Valley (France)

PONE-D-21-37152R1

Dear Dr. ALLARD,

We’re pleased to inform you that your manuscript has been judged scientifically suitable for publication and will be formally accepted for publication once it meets all outstanding technical requirements.

Kind regards,

John P. Hart, Ph.D.

Academic Editor

PLOS ONE

Additional Editor Comments (optional):

Reviewers' comments:

Reviewer's Responses to Questions

**Comments to the Author**

1. If the authors have adequately addressed your comments raised in a previous round of review and you feel that this manuscript is now acceptable for publication, you may indicate that here to bypass the “Comments to the Author” section, enter your conflict of interest statement in the “Confidential to Editor” section, and submit your "Accept" recommendation.

Reviewer #2: All comments have been addressed

2. Is the manuscript technically sound, and do the data support the conclusions?

Reviewer #2: Yes

3. Has the statistical analysis been performed appropriately and rigorously? 

Reviewer #2: Yes

4. Have the authors made all data underlying the findings in their manuscript fully available?

Reviewer #2: Yes

5. Is the manuscript presented in an intelligible fashion and written in standard English?

Reviewer #2: Yes

6. Review Comments to the Author

Reviewer #2: Re-evaluating the article under review, I would like to once again emphasize the high professionalism of the authors, the significant contribution of their methodological developments to socio-economic research based on the study of the features of ancient technologies for processing flint raw materials and manufacturing blade blanks. A thorough technological analysis of flint products from several archaeological objects of the LPC culture allowed the authors to show that the details of flint knapping characterize not only the technological traditions of this stage of cultural development, but can also be a source for interpreting the individual characteristics of flint raw materials processing both in different residential complexes and in the same dwelling.

Reworked by the authors, taking into account the comments of the reviewers, the article has acquired a more reasoned basis for the assumptions made. In the "methods and materials" section, information about the archaeological context has been significantly expanded. In the “Results” section, data on the nature of local flint raw materials are supplemented, and the review of experimental works is also significantly expanded, which allowed the authors to better argue the statements made about the nature of flint knapping technology. The extended section "Conclusions" has become a full-fledged summary of the study, showing the wide interpretive possibilities of a detailed study of the features of ancient technology.

Thus, the peer-reviewed article can be published. It is certainly an interesting study based on an original technique for studying flint artefacts, which can initiate a wide discussion of the results among specialists.

I wish the authors further success in their work, successful completion of the LPC flint industry research project, and a wider application of the proposed methodology to other Neolithic archaeological sites.

7. PLOS authors have the option to publish the peer review history of their article (what does this mean?). If published, this will include your full peer review and any attached files.

Reviewer #2: No

---

## [Editor Report · Acceptance letter]

10 May 2022

PONE-D-21-37152R1 

Technical traditions and individual variability in the Early Neolithic: Linear Pottery Culture flint knappers in the Aisne Valley (France) 

Dear Dr. ALLARD:

I'm pleased to inform you that your manuscript has been deemed suitable for publication in PLOS ONE. Congratulations! Your manuscript is now with our production department. 

Kind regards, 

on behalf of

Dr. John P. Hart 

Academic Editor

PLOS ONE